# Construction and application of a novel urban knowledge model with extended historical and cultural semantics

**Chaoqun Wang**[1], **Cuicui Xu**[1‡], **Yinglin Wang**[1], **Jie He**[1*], **Weijiang Pan**[2], **Xin Xu**[2]

**1** School of Architecture, Harbin Institute of Technology, Shenzhen, Shenzhen, Guangdong Province, China, **2** The 10th Institute of Architectural Design, Guangdong Architectural Design and Research Institute Group Co., Ltd., Guangzhou, Guangdong Province, China

‡ Deceased
* hejie2021@hit.edu.cn

**Data availability statement:** The data and code is available on Github: https://github.com/DaqunW/UrbanKG_GZ_case.

## Abstract

Historical urban spaces, imbued with profound historical and cultural significance, have evolved from 'spaces' into meaningful 'places'. But they now face the risk of being eroded by rapid urbanisation and forgotten by today's society, making it challenging to integrate them into modern life. UNESCO's Historic Urban Landscapes (HUL) Recommendation highlights the importance of understanding the holistic, layered, and dynamic nature of urban heritage. Therefore, focusing on the ancient city district of Guangzhou as a case study, this research explores innovative approaches to integrating holistic semantics of urban places by merging contemporary semantics from geo-big data with historical and cultural semantics from documents and archives to create an urban knowledge model that bridges the gap between the present and the past. Furthermore, using knowledge graph embedding technology, we develop a model capable of entity prediction, similarity calculation, and query retrieval. We propose four key application scenarios for the implementation of the model. First, our research identifies potential cultural spatial connections that contribute to the joint preservation and promotion of historic urban places. Second, we develop a recommendation system that caters to users' various requests, increasing the visibility of historical places. Third, we predict optimal locations for Time-Honored Brands. Finally, we identify visitor profiles to assist managers in meeting cultural promotion needs. To summarize, the integrated framework proposed in this study demonstrates both methodological efficacy and reusability. It not only helps to deeply explore the historical and cultural connotations, providing a scientific basis for urban planning and cultural inheritance, but also has the potential for enhancing the public's awareness and participation in historical culture, promoting the sustainable development and prosperity of urban culture.

## Introduction

Urban space, as a medium of human activities, memories, and emotions, is transformed from "space" into "place" through affective attachments and associated meanings [1–3].

**Funding:** Research Initiative Fund for Newly Introduced Talents of Harbin Institute of Technology, Shenzhen (#ZX20230488).

**Competing interests:** The authors have declared that no competing interests exist.

The information embedded within such places is termed *place semantics*. The perception of place semantics directly influences urban cultural identity, community cohesion, and the formation of a sense of place [4] all of which are crucial for social innovation, cultural vitality, and sustainable urban development [5]. Place semantics comprises two dimensions, contemporary semantics and historical-cultural semantics. Contemporary semantics are continually shaped by present lived experiences and immediate socio-spatial interactions. In contrast, historical and cultural semantics are constituted and accumulated through prolonged processes of temporal stratification and urban evolution [6], encompassing cultural elements, historical events, notable figures, and collective practices. However, the historical and cultural semantics of places are facing unprecedented challenges. The combined forces of globalisation, digital communication paradigms, and China's rapid economic transformation have fundamentally reconfigured urban landscapes and living conditions. Consequently, Contemporary people's perceptions are shaped more by information society behaviours, while the memories related to historical and cultural semantics have faded.

In response to these pressing challenges, UNESCO has advocated for a more comprehensive understanding of urban cultural heritage values through the publication of key documents such as the *Vienna Memorandum* (2005) [7] and the *Historic Urban Landscape (HUL) recommendation* (2011) [8]. The HUL recommendation emphasizes the importance of recognizing the diverse values of urban cultural heritage, particularly in the context of the relationships between people, places, and temporal change, underscoring the importance of this heritage for society [9,10]. It also emphasizes the need for a holistic and systematic approach to the conservation of urban cultural heritage, which requires an accurate understanding of the spatiotemporal continuity of these heritage sites [11–14]. Concurrently, the advent of digital technologies has catalysed transformative approaches within the digital heritage domain. UNESCO's 2003 *Charter on the Preservation of the Digital Heritage* advocates the systematic digitisation and preservation of culturally significant historical materials as enduring public assets [15,16]. In recent years, governments and other institutions have made efforts to hold symposiums or exhibitions and open document databases to the public, aim to bridge the gap between the public and urban historical memory. However, the knowledge that the public can access through these channels is fragmented, localised, and disjointed.

Recent scholarly research has often explored either the historical-cultural or the contemporary semantics independently. Artificial intelligence (AI) technology has made significant contributions to both aspects. On one hand, in studies related to the historical and cultural semantics, symbolic AI has facilitated the development of numerous open-source knowledge graph projects [17], as well as various urban memory projects [18]. On the other hand, in the field of urban informatics, connectionist AI, particularly machine learning, has driven continuous innovation in spatial semantic extraction. It regards place semantics as the process of extracting the physical attributes of spaces and elements within them, such as the spatial functions, visual features, and characteristics of crowd behaviour, which serve as inputs for representation learning. The results of representation learning are applied to the identification of spatial similarities in urban areas and the delineation of functional zones. However, little attention has been paid to the extraction and integration of the dual-dimensional semantics concurrently, resulting in a lack of continuous and comprehensive representation of urban place semantics.

In summary, the historical and cultural dimension of place semantics currently faces the risk of fading into public oblivion, underscoring the urgent need for a comprehensive and systematic analysis of its spatiotemporal development process. Digital methodologies and artificial intelligence demonstrate significant potential in revitalising place historical and cultural semantics. This study addresses the core research question: How can digital technologies

construct an effective bridge between historical-cultural and contemporary place semantics? We argue that resolving this question requires a holistic digital infrastructure capable of integrating the dual-dimensional semantics of urban places, encompassing both contemporary and historical-cultural dimensions. Furthermore, practical implementation pathways must be outlined to maximise the potential of such an infrastructure.

Our contribution is the development of a holistic methodological framework that encompasses the construction of a knowledge graph integrating both contemporary and historical-cultural semantics, the training of embedding models, and their application scenarios. The viability of this framework is empirically demonstrated through a case study conducted in Guangzhou.

## Literature review and research gaps

### Extracting historical and cultural semantics in the digitisation of cultural heritage

Archives and library and information science, the imperative for cultural asset management has spurred a multitude of research initiatives on the digitisation of cultural heritage. These endeavours commonly leverage ontologies, semantic web technologies, and knowledge graphs to organise semantic knowledge effectively, represent it formally, and utilise it in a standardised manner. Some of these projects are dedicated to providing standardised databases. For example, the standardised vocabulary knowledge graphs for Chinese historical figures, historical eras, and places' names created by the Shanghai Library [19]. In addition, some thematic knowledge graphs are designed to structure knowledge within a particular domain [20–30]. Building upon this, visual platforms [18,31–33,35] or apps [34] tailored for the general public are developed. Several open-source projects currently demonstrate the capacity to integrate dual-dimensional place semantics, despite their primary focus on extracting historical-cultural semantics. The Pelagios [36] project focuses on the open storage and linking of historical geographical information pertaining to the ancient Mediterranean and adjacent regions. It supports comparative visualisation between locational data and semantic place networks. It has developed Recogito [37], a tool enabling manual annotation of geographical, personal and event-related semantics in texts and images – thereby significantly enriching the project's historical-cultural semantic layer. Similarly, the World Historical Gazetteer (WHG) [17] catalogues global historical places toponyms across extended temporal sequences, allowing people to make connections across time, space and language. The Linked Places [38] project offers even richer semantic connections: locations may be linked both through modern physical infrastructure (e.g. roads and railways) and through historically attested routes (e.g. the Silk Road). The digitisation of historical semantic dimensions critically depends on detailed semantic annotation of multimedia cultural resources [39–41].

Recently, governments and heritage institutions are increasingly adopting digital innovations for heritage conservation. By adopting emerging digital technologies such as augmented reality (AR) and deep mapping [42], these organisations have developed a range of digital narrative projects using urban historical-cultural spaces as medium [43], while actively incorporating such initiatives into urban planning practices [44]. These projects employ diverse narrative strategies [43,45,46] to reveal hidden cultural meanings in urban spaces, improving public understanding of historical and cultural semantics and community engagement in heritage preservation [43]. It has spurred numerous geo-tagging enabled narrative tools [47] (such as StoryMap [48], TimeMapper [49] and Odyssey [50], etc.). These projects' digital infrastructure is mostly spatial database systems, which exhibit significant limitations. On the one hand, it is ill-equipped to efficiently store and manage relational semantic data, thereby

limiting reasoning capabilities; on the other hand, it lacks the capacity to support large-scale, heterogeneous knowledge systems.

### The semantic extraction of places with the support of spatial big data

From the early stages of human geography to the rapidly advancing field of urban computing, there has long been a focus on the extraction and analysis of semantics. Human geography posits that 'place' refers to spaces that are associated with human consciousness, emotions, or behavioural experiences [1]. Therefore, the current research on spatial semantics can be more accurately described as an exploration of the semantics of places. Wang S [51] categorised the place semantics characterised by spatio-temporal big data into semantic and associative attributes. Studies focusing on semantic attributes typically rely on extracting patterns of place activities and their temporal variations [52–57], emotions [58–60], and functional semantics [61–65]. In contrast, research on associational attributes focuses on quantifying the interaction intensity between places [68–75]. Subsequently, more researches tend to develop spatial embedding methods by combining multimodal datasets [76,77] to capture spatial features and then integrate them, incorporating POI [78,79], trajectory OD [80], temporal crowd flow, spatiotemporal movement patterns [81], social media texts [82], street view imagery [83], urban texture images [84,85], and road network topology [86]. The embedding results can further facilitate tasks such as calculating place similarity, determining place functions [87], recommending locations based on user behavior preferences [88,89], spatial units' community detection [66,67], and predicting urban context attributes of interest to other researchers [90,91]. Liu Yu's team integrated such studies to develop the UrbanKG framework [92], which interprets the entities and relationships depicted by the spatial big data commonly used in research, can aggregate multi-source urban semantics. [93–96].

### Research gaps

Historical-cultural place semantics extraction studies have successfully employed knowledge graphs to structure and visualise urban historical semantics. Several open-source projects enhance public accessibility to historical place semantics and inter-place relationships through visualisation and query functionalities. However, these initially structured semantic representations remain rudimentary, with latent relational patterns in the data yet to be fully exploited. Conversely, contemporary place semantics extraction studies demonstrate methodological sophistication but disregard the history. The application of deep learning in place semantics extraction and downstream tasks [92] suggests a promising approach: by structurally organising historical semantics annotations of cultural resources via a knowledge graph, then integrating these with deep learning embeddings, we can develop intelligent services that enable computationally tractable and interpretable fusion of dual-dimensional place semantics.

## Data and methods

In this section, we introduce the study area and explain the technical approach of this paper step by step.

### Study area

The study focuses on the ancient city district of Guangzhou (as shown in Fig 1), including the Liwan, Yuexiu, and Haizhu districts. Renowned for its rich historical sites and cultural

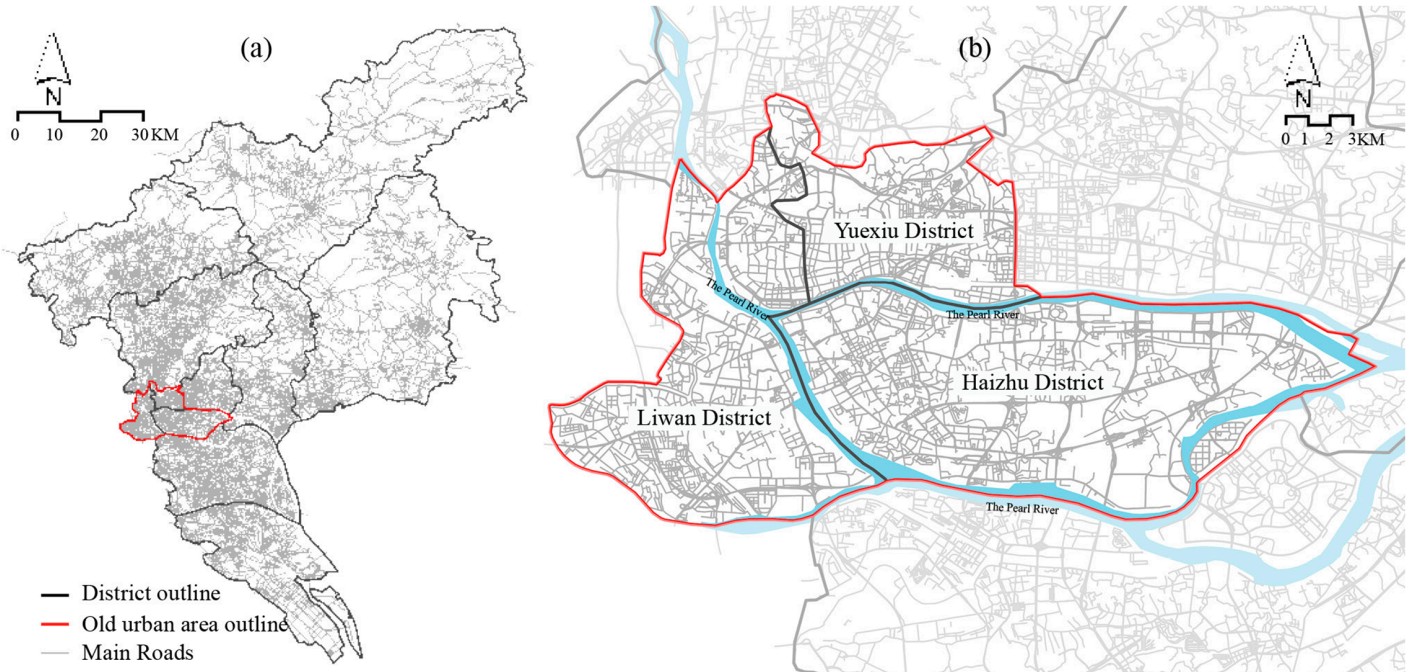

**Fig 1. Location of study area.** (a) The location of the ancient city district of Guangzhou (b) The map of the ancient city district of Guangzhou. Contains information from OpenStreetMap and OpenStreetMap Foundation, which is made available under the Open Database License.

heritage, Guangzhou is a melting pot of Red culture, Lingnan culture, trade culture as represented by the Maritime Silk Road, and other innovative cultures. Due to its key position in the Maritime Silk Road, Guangzhou has thus earned its reputation as the "Millennium Commercial Capital." Since the Qin Dynasty, Guangzhou has served as the administrative centre of the Lingnan region of ancient China. In the modern era, Guangzhou has been a pivotal city for China's bourgeois democratic revolution and proletarian struggles. For over 2,200 years, the city has maintained its original site, and its 20.39 square kilometres of historical areas preserve a rich urban fabric and cultural resources, including numerous historical and cultural neighbourhoods, buildings, and traditional villages.

## Methodology

We propose an approach consisting of five components, as illustrated in Figure 2. This study collects multi-source urban data across contemporary and historical dimensions, which undergoes spatial alignment and is aggregated into 100m × 100m grid cells. Subsequently, the entity and relationship types within the dataset are analysed to define a schema encompassing knowledge from both contemporary and historical dimensions. The data is then transformed into triplets to construct an urban knowledge graph. Next, the triplets are input into a Knowledge Graph Embedding (KGE) model to learn vector representations of entities and relations. Based on these embeddings, three key functions are developed: Link prediction (including head/tail entity prediction), Cosine distance computation, and Similarity measurement. Leveraging these functions, four application scenarios are proposed: Mining potentially similar locations and entities, Destination recommendation, Site selection analysis for

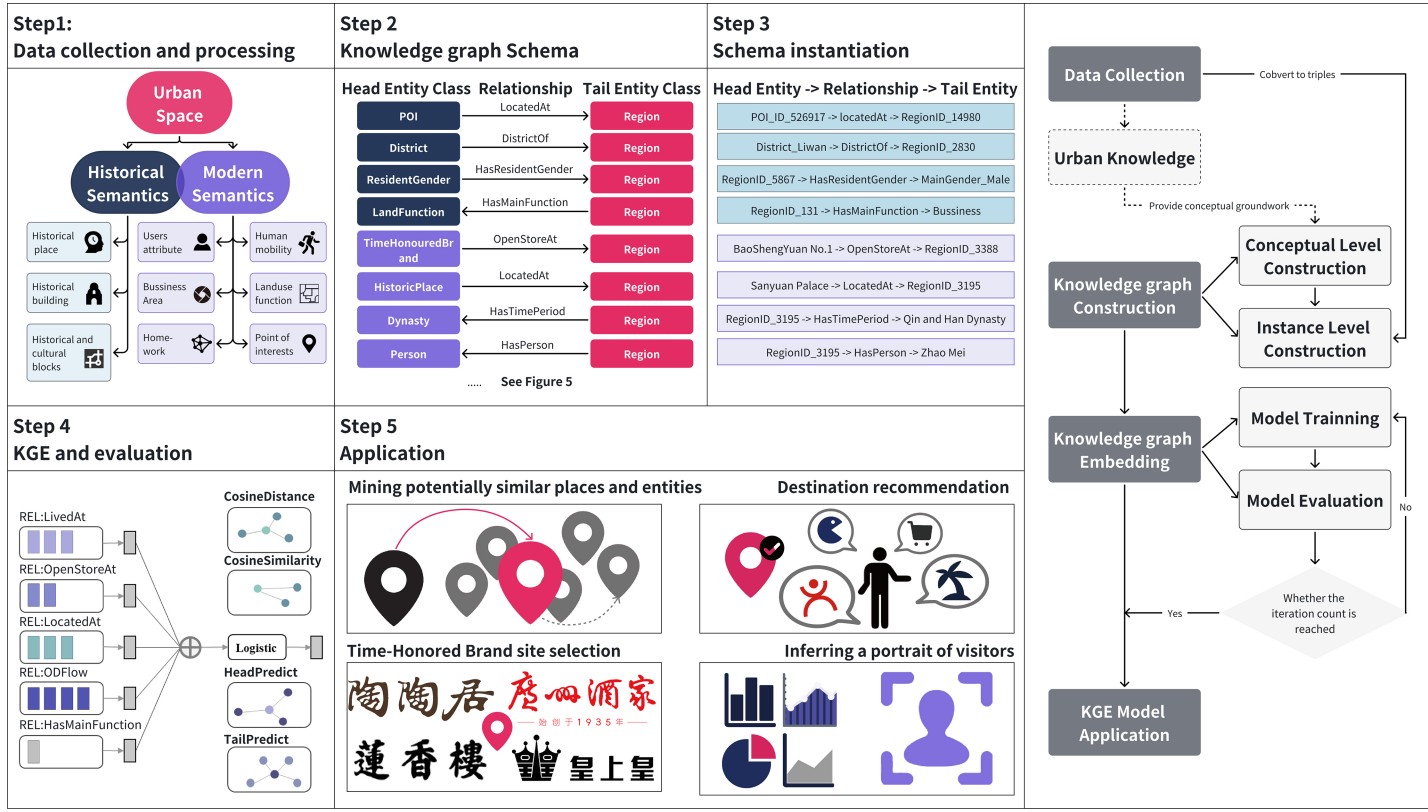

**Fig 2. Methodology flowchart.**

time-honoured brands, and Visitor profile inference. Together, these components establish a comprehensive framework for urban knowledge model construction and application.

**Data collection and processing**

**Basic units and contemporary data processing.**

Contemporary data, predominantly sourced from the Baidu API (which uses **100 × 100 m grids** as its smallest sampling unit), also constitute the fundamental spatial units of our study. The contemporary data are categorised into two types, as follows.

(1) Grid Attributes, include the demographic and occupational profiles, POIs, land use functions (calculated with reference to [97]), and the number of Weibo check-ins within each grid. The vitality level of each grid is also assessed, as detailed in Section *Cultural level and vitality level evaluation*.

(2) Inter-grid connections refer to the interactions between grids, incorporating OD flow data, commuting data, and visitation relationships between external populations and grids within the study area.

**Unstructured historical data processing.**

Historical Data originates from the digitisation and structural processing of various ancient texts, maps, and documents.

(1) Spatial elements, include key historical sites in urban history, historical and cultural districts, and historical buildings reflecting urban historical landscapes. Each record corresponds to a record in the GIS spatial layers, encompassing point, line, and polygon types, as shown in Fig 3. Then they connect to spatial grids using a spatial join tool.

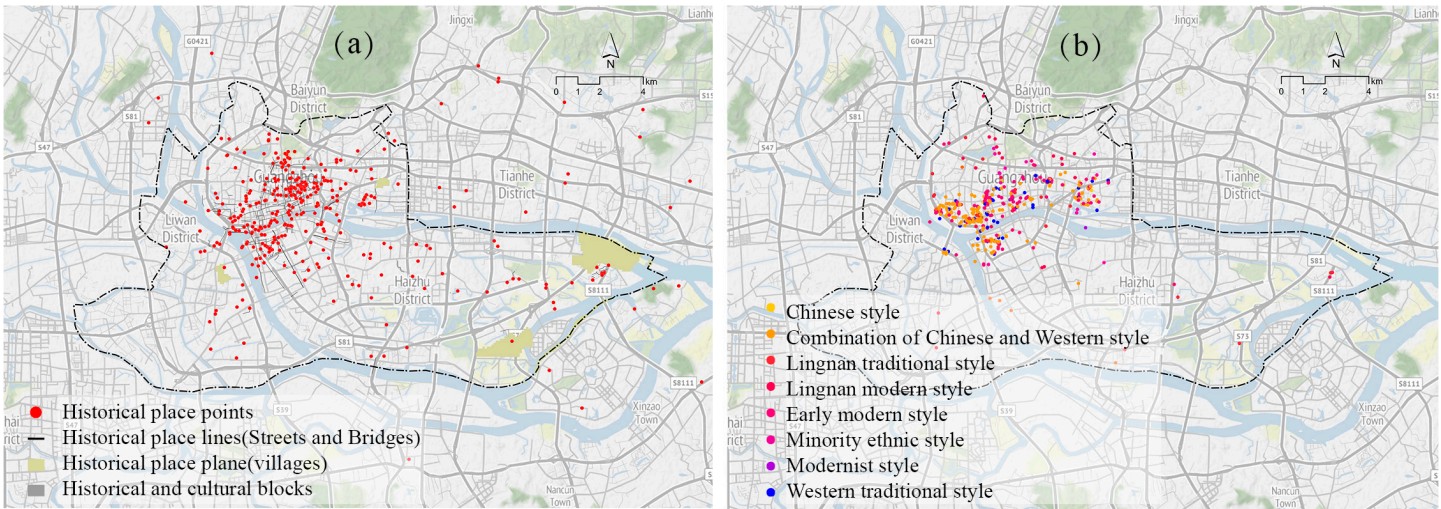

**Fig 3. Spatial distribution of historical data.** (a) Historical sites and historical cultural blocks; (b) Historical buildings. Image courtesy of the Earth Science and Remote Sensing Unit, NASA Johnson Space Center

(2) Other elements mainly include related historical figures, cultural types and time periods associated with these spatial entities, are also recorded into GIS database. The cultural type was obtained through manually tagging the records. This study uses the grounded theory [98] workflow to develop a cultural tagging system. This cultural tagging system employs a structured taxonomy to classify and describe historical sites and provides essential annotations for their historical and cultural semantics. Its tags interconnect through both hierarchical and associative semantic relationships. As shown in Fig 4. Each record is tagged with relevant cultural labels, and multiple tags are allowed for each record. In addition, time periods are divided into several segments based on significant historical events, as detailed in Table 1.

The extraction of attribute information in historical data processing relies heavily on manual interpretation. The raw unstructured historical materials used in this study typically contain explicit spatial information (e.g., locations of historical places or buildings), historical figures, and events elements, making these features relatively straightforward to identify. However, the classification of cultural types based on the cultural tagging system involves subjective judgment, potentially leading to discrepancies among different annotators. To ensure the rationality and standardization of cultural tag assignments, this study implemented a quality control procedure consisting of three stages: initial parallel annotation by multiple annotators, followed by cross-validation, and finalised by a dedicated reviewer for verification.

**Schema of the urban knowledge graph: Bridging historical-cultural and contemporary semantics.** Focusing on our basic spatial units, we extract key elements from all of the data as entities and consider the multidimensional relationships between these entities, which involve their spatial and semantic relationships. We thus develop an urban knowledge schema that integrates both historical and contemporary semantics (as illustrated in Fig 5).

The construction process of the schema takes into account the following factors:

- Historical and cultural semantic dimension: Capturing entities from the perspectives of historical figures, places, activities, and events.

| ID | Type | Dynasty | Name | Events | Completion time | Disappearance time | Cultural label | | Overview | Detail |
|---|---|---|---|---|---|---|---|---|---|---|
| 1 | Scenic spots and historical sites | Qing | Sifang fort barbette | Before the Opium Wars | 1653 | present | City defense | | Opium Wars | Located within the Yuexiu …. |
| 2 | Scenic spots and historical sites | Modern era | Guangzhou Sacred Heart Cathedral | Opium Wars to the Hundred Days' Reform | 1863 | present | Religious (Catholicism) | | It is named for its granite construction and is a Gothic-style building. | Located in the Jinghai …. |
| 3 | Scenic spots and historical sites | Modern era | Guangzhou Christian Dongshan Church | Opium Wars to the Hundred Days' Reform | 1870 | present | Religious (Christianity) | | Named after its location in Dongshan and its affiliation with Christianity | Located in the Sibei…. |

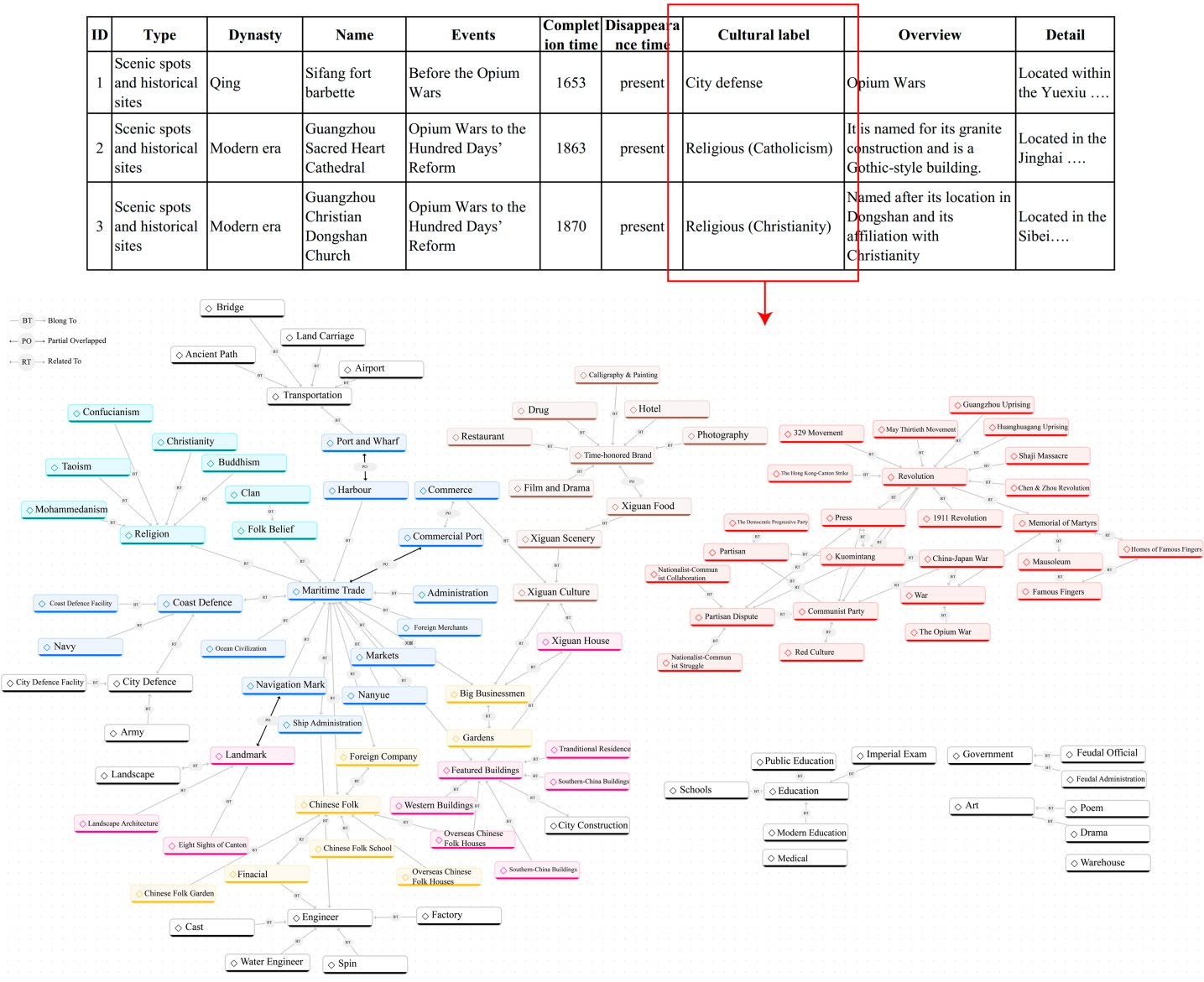

**Fig 4. Historical data cultural labels.** (a) Historical data table records; (b) Cultural tagging system

- Contemporary semantic dimension: Identifying not only the basic attributes of space, such as function and job or housing attributes, but also capturing structural relationships like topology and interaction.
- Standardised node names: Ensuring easy linkage with external knowledge graphs. For instance, the names of all individuals under the *Person* class should be consistent with those in the name authority database of the Shanghai Library [99].
- High reusability frequency: Entities and relationships should possess a high frequency of reuse.

**Table 1. Segmentation of time period.**

| Events | Temporal division |
|---|---|
| Before the Opium Wars | 1644–1840 |
| Opium Wars to the Hundred Days' Reform | 1840–1898 |
| Hundred Days' Reform to the Establishment of the Republic of China | 1898–1911 |
| Establishment of the Republic of China to the May Fourth Movement | 1911–1919 |
| May Fourth Movement to the Fall of Guangzhou | 1919–1938 |
| Fall of Guangzhou to the Victory in the War of Resistance Against Japan (Second Sino-Japanese War) | 1938–1945 |
| Chinese Civil War to the Founding of the People's Republic of China | 1945–1949 |
| After the Founding of the People's Republic of China | 1949– |

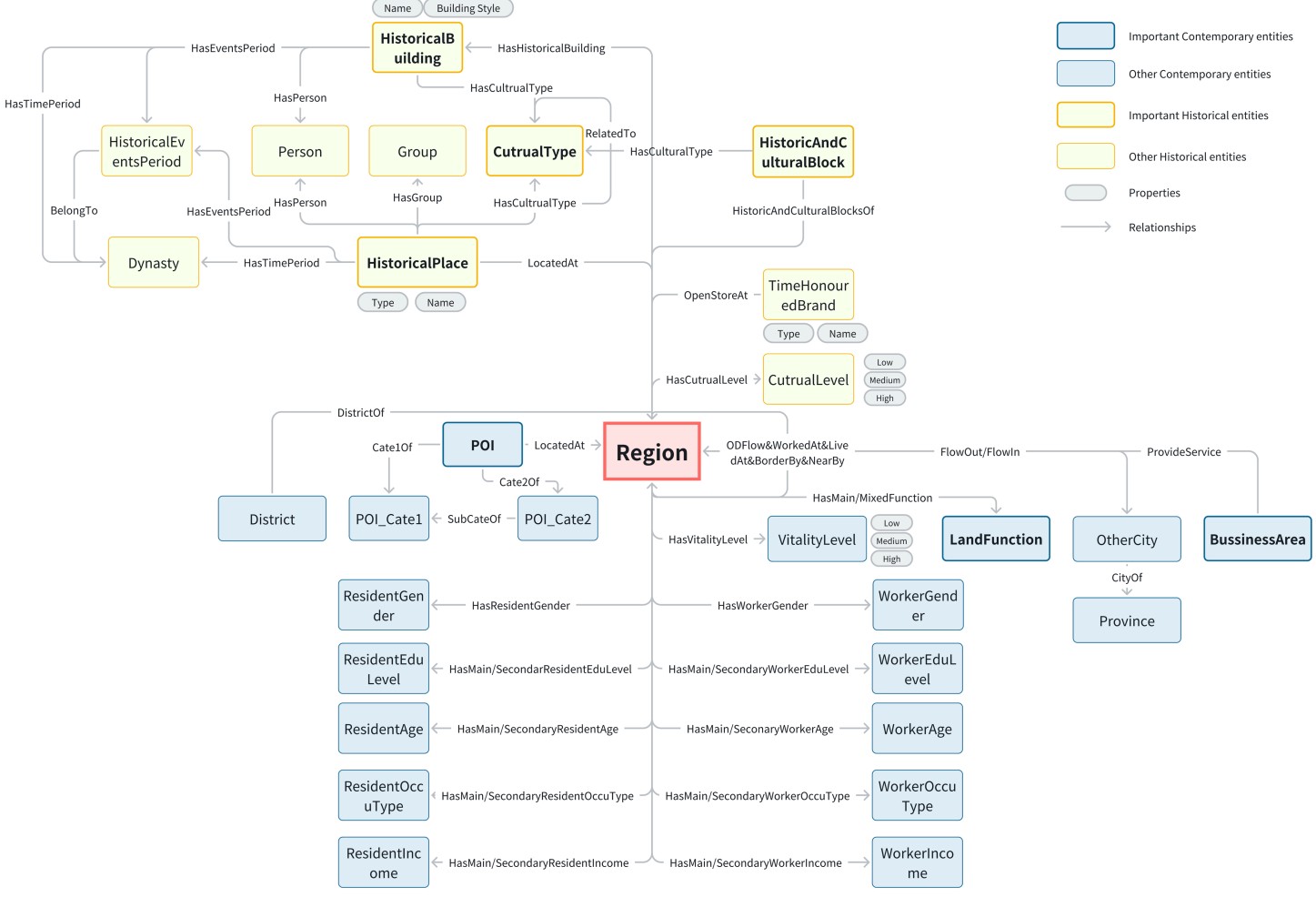

**Fig 5. The schema of the urban knowledge graph.**

The urban knowledge graph with the *Region* class is the fundamental spatial unit. The upper section represents the historical semantic layer, while the lower section represents the contemporary semantic layer.

The definitions of the entities and relationships are provided in Tables 2 and 3.

**Table 2. Summary of entities and required data.**

| Entity name | Semantics | Source |
|---|---|---|
| HistoricalBuilding | Historical buildings designated by Guangzhou's Cultural Heritage Census | *List of Historical Buildings in Guangzhou (1-7)* |
| Person | Famous historical figures, such as the King of Nanyue | / |
| Group | Renowned historical groups, such as the Xingzhong Society | / |
| CulturalType | Cultural tags; see Fig 4 for details | / |
| HistoricalPlace | Historical Place Names in Guangzhou | *Guangzhou Place Name Protection Catalogue* |
| HistoricAndCulturalBlock | Guangzhou's Historical and Cultural Districts | *Guangdong Province List of Historical and Cultural Blocks* and their conservation planning documents |
| Dynasty | Dynasty of China | / |
| HistoricalEventsPeriod | The period from the late Qing Dynasty to the Republic of China is further divided into significant time periods; see Table 1 for details | / |
| TimeHonoredBrand | Long-standing and reputable brands or enterprises | *Time-Honored Brands Recognition Director* |
| CulturalLevel | Grid's level of historical and cultural atmosphere | calculation detail in Section *Cultural level and vitality level evaluation* |
| Region | Research area divided into 100m × 100m grids based on Baidu data | Spatial grids from Baidu data |
| POI | Points of interest | POI from Amap API |
| POI_Cate1 | Major categories of POI | ibid. |
| POI_Cate2 | Subcategories of POI | ibid. |
| District | Administrative divisions | / |
| LandFunction | Land use | Calculated with reference to [97] |
| OtherCity | Cities in other provinces with population inflow or outflow from the study area | Population mobility data from Baidu data |
| Province | Other provinces with population inflow or outflow from the study area | / |
| BussinessArea | Business districts | *Guangzhou Key Commercial Functional Zone Development Plan (2020-2035)* |
| VitalityLevel | Grid's level of vitality | calculation detail in Section *Cultural level and vitality level evaluation* |
| ResidentGender | Gender characteristics of permanent residents within the grid, with three values: male-dominant, female-dominant, and balanced | Population portrait from Baidu data |
| ResidentEduLevel | Educational level of permanent residents within the grid, with three values: Bachelor's degree and above, Associate's degree, and high school and below | ibid. |
| ResidentAge | Age of permanent residents within the grid, with three values: youth (0–18 years), middle age (35–54 years), and elderly (55 years and above) | ibid. |
| ResidentOccuType | Types of work of permanent residents within the grid, with six values: production operations, clerical, professional technical, management, self-employed, and service staff | ibid. |
| ResidentIncome | Income of permanent residents within the grid, divided into five values: Below 2,500 CNY, 2,500 CNY–3,999 CNY, 4,000 CNY–7,999 CNY, 8,000 CNY–19,999 CNY, and 20,000 CNY and above | ibid. |
| WorkerGender | Gender characteristics of the working population within the grid, with three values: male-dominant, female-dominant, and balanced | ibid. |
| WorkerEduLevel | Educational level of the working population within the grid, with three values: Bachelor's degree and above, Associate's degree, and high school and below | ibid. |
| WorkerAge | Age of the working population within the grid, with three values: youth (0–18 years), middle age (35–54 years), and elderly (55 years and above) | ibid. |
| WorkerOccuType | Types of work of the working population within the grid, with six values: production operations, clerical, professional technical, management, self-employed, and service staff | ibid. |
| WorkerIncome | Income of the working population within the grid, divided into five values: Below 2,500 CNY, 2,500 CNY–3,999 CNY, 4,000 CNY–7,999 CNY, 8,000 CNY–19,999 CNY, and 20,000 CNY and above | ibid. |

**Table 3. Summary of relations and corresponding semantics captured.**

| Relation | Head entity | Tail entity | Semantics |
|---|---|---|---|
| HasHistoricalBuilding | Region | HistoricalBuilding | The historical building is located within this region |
| HasPerson | HistoricalBuilding/ HistoricalPlace | Person | The building or location is associated with a historical figure |
| HasGroup | HistoricalPlace | Group | The location is associated with a historical group |
| HasCulturalType | HistoricalBuilding/ HistoricalPlace/ HistoricAndCulturalBlock | CulturalType | Define the cultural types of the location, historical building, or historical and cultural district |
| BelongTo | HistoricalEventsPeriod | Dynasty | Major events occurred during this dynasty |
| HasTimePeriod | HasHistoricalBuilding/ HistoricalPlace | Dynasty | The historical site or building appeared during this dynasty |
| HasEventsPeriod | HistoricalBuilding/ HistoricalPlace | HistoricalEventsPeriod | The historical site or building appeared during this time period |
| HistoricAndCultural BlocksOf | HistoricAndCulturalBlock | Region | The area is located within this historical and cultural district |
| OpenStoreAt | TimeHonoredBrand | Region | TimeHonored Brands have stores in this area |
| HasCulturalLevel | Region | CulturalLevel | The rating of the historical and cultural atmosphere of this area |
| ODFlow | Region | Region | According to Baidu Huiyan's OD data, there are more than five people moving between region A and region B |
| WorkedAt | Region | Region | More than 5 people live in region A and work in region B |
| LivedAt | Region | Region | More than 5 people work in region A and live in region B |
| BorderedBy | Region | Region | The area A and area B grids are connected by at least one edge |
| NearBy | Region | Region | The A grid is within 500 meters of the B grid |
| LocatedAt | POI | Region | The POI is located within this area |
| Cate1Of | POI_Cate1 | POI | Used to describe the major category to which the POI belongs |
| Cate2Of | POI_Cate2 | POI | Used to describe the subcategory to which the POI belongs |
| DistrictOf | District | Region | Describes the administrative division of the region |
| HasMainFunction | Region | LandFunction | Refers to the main function of the area |
| HasMixedFunction | Region | LandFunction | Refers to the secondary function of the area |
| FlowOut | Region | OtherCity | More than 5 people flow from this area to the city |
| FlowIn | OtherCity | Region | More than 5 people flow from the city to this area |
| CityOf | OtherCity | Province | Provinces where cities with population interaction in the study area are located |
| ProvideService | BusinessArea | Region | The area is within this business district |
| HasVitalityLevel | Region | VitalityLevel | The vitality rating of the area |
| HasResidentGender | Region | ResidentGender | The gender characteristic value of the resident population in the area |
| HasMain/SecondaryResidentEduLevel | Region | ResidentEduLevel | The education level value of the resident population in the area |
| HasMain/SecondaryResident entAge | Region | ResidentAge | The age characteristic value of the resident population in the area |
| HasMain/SecondaryResidentOccuType | Region | ResidentOccuType | The type of work characteristic value of the resident population in the area |
| HasMain/SecondaryResidentIncome | Region | ResidentIncome | The income value of the resident population in the area |
| HasWorkerGender | Region | WorkerGender | The gender characteristic value of the working population in the area |
| HasMain/SecondarWorker EduLevel | Region | WorkerEduLevel | The education level value of the working population in the area |
| HasMain/SecondaryWorker Age | Region | WorkerAge | The age characteristic value of the working population in the area |
| HasMain/SecondaryWorker OccuType | Region | WorkerOccuType | The type of work characteristic value of the working population in the area |
| HasMain/SecondaryWorker Income | Region | WorkerIncome | The income value of the working population in the area |

**Ontology instantiation.** The Ontology instantiation process reorganises the data from the *entity, attribute1, attribute2, etc.* format into a structure that conforms to the triple (*head entity-relation-tail entity*) format. There is an example:

- *POI_ID_1, Cate1Of, POI_Cate1_TransportationInfrastructure*
- *POI_ID_1, Cate2Of, POI_Cate2_BusStation*
- *BusStation, SubCateOf, TransportationInfrastructure.*

**Knowledge graph embedding and model evaluation.** We use KGE technology to obtain the vectorised representations of all of the triples obtained from the previous step. Based on the vectorised representations, we further develop functions that include cosine distance and similarity computation, link prediction(head or tail entities prediction). We split the knowledge graph into subgraphs to train three KGE models, one considering only contemporary semantics (M1), one considering only historical and cultural semantics (M2), and a combined model considering both contemporary and historical semantics (M3). The training parameters of all three models are compared. The approach of decoupling and training distinct dimensional semantics separately facilitates flexible model selection tailored to specific application scenarios.

**Application of knowledge models.** We propose four application scenarios for the urban knowledge model, implemented through downstream tasks such as prediction and similarity computation. We hope these scenarios can uncover intrinsic connections between historical and contemporary dimensions of place semantics, thereby enhancing public engagement with cultural heritage in daily life; and provide valuable insights for urban planning policy formulation and cultural heritage conservation efforts. Guangzhou's Liwan Historical and Cultural District is chosen as the main empirical case to practically validate the effectiveness of the models and to demonstrate the implementation process of our knowledge model empowerment pathways.

## Main algorithm

The main algorithms utilised in this paper are as follows:

**Knowledge graph embedding (KGE).** This paper selects the ConvE model from convolutional neural networks for graph embedding and prediction. ConvE is an algorithm specifically developed for knowledge graph embedding, demonstrating significant advantages in classical tasks such as link prediction, which aligns well with the requirements of this study. By leveraging its unique local convolution operations, ConvE can efficiently learn the interaction patterns between entities and relations, making it suitable for modelling complex semantic relationships, including many-to-many and one-to-many mappings. ConvE has been empirically proven to outperform traditional knowledge graph embedding models such as TransE, DistMult, and ComplEx in link prediction tasks [100]. Although RotatE [101,102] and CompGCN [103] exhibit superior performance when handling large-scale datasets and complex relations, this study ultimately adopts ConvE due to the single-city scale of the dataset and the consideration of model lightweight requirements.

We assume that a knowledge graph $G = \{(s,r,o)\} \subseteq \varepsilon \times R \times \varepsilon$, $\varepsilon$ is the entity set and $R$ is the relationship set. The objective of the model is to learn a score function. Given an input triple $x=(s,r,o)$, its score function $\psi(x)$ is the likelihood that the fact encoded by triple $x$ is true. The score of the triple $(s,r,o)$ is defined as $\psi_r(e_s, e_o)$:

$$\psi_r(e_s, e_o) = f(vec(f([\overline{e_s}; \overline{r_r} * \omega])W)e_o \tag{1}$$

$f$ represents the activation function $Relu()$. $vec()$ represents the full connection. $e_s$ and $e_o$ denote the head entity and tail entity embedding, respectively. $r_r$ is a relation parameter depending on r. $\overline{e_s}$ and $\overline{r_r}$ denote 2D reshapings of $e_s$ and $r_r$, respectively. $W$ is the weight matrix. $\omega$ is the convolution kernel.

**Model training and performance evaluation.** We divide the input data to develop three distinct models. During the training of each model, the dataset was divided into training, cross-validation, and test sets with a ratio of 8:1:1. The learning rate was set to 0.001, and the

models were trained for a total of 100 epochs. We calculate the following metrics to evaluate the model performance:

(1) Mean reciprocal rank (*MRR*)

*MRR* is a metric used to evaluate the effectiveness of retrieval systems. It measures the average of the reciprocal ranks of the first relevant item in the search results. Essentially, MRR assesses how quickly a system returns a correct answer in a series of queries, with higher MRR values indicating better performance.

$$MRR = \frac{1}{Q} \sum_{i=1}^{|Q|} \frac{1}{rank_i} \tag{2}$$

$|Q|$ is the number of all candidates and $rank_i$ is the right answer's rank in the reference list for the *ith* item.

(2) HITS@n

HITS@n measures the proportion of times that a relevant item appears in the top-n ranked results returned by the model. For example, if HITS@10 is 0.8, it means that 80% of the time, a relevant item is ranked in the top 10 results.

$$HIT@n = \frac{1}{Q} \sum_{i=1}^{|Q|} (rank_i \leq n) \tag{3}$$

The meanings of the variables in the above formula are the same as in Equation (2).

**Graph database evaluation and visualisation.** This study uses Neo4j to evaluate and visualise our knowledge database. Neo4j is a high-performance graph database capable of transforming data into a graph format that is easy to understand and analyse, making it well-suited for user interaction [104]. In Neo4j, queries can be executed using the Cypher query language.

Additionally, the GDS (Graph Data Science) plugin of Neo4j provides algorithms for graph network analysis. In this study, we use the average density, average degree centrality, and average closeness centrality to analyse our knowledge graph network.

The average density is defined as the ratio of the number of instances in the knowledge base to the number of classes defined in the ontology schema. A higher density indicates that the knowledge base contains richer facts or knowledge.

For node *i* in graph *G*, the degree centrality of node *i* refers to the number of edges incident to it. Therefore, the average degree centrality is the mean of the degree centralities of all of the nodes.

The closeness centrality of node *i* is calculated by

$$C_i = \frac{1}{N-1} \sum_{j \neq i} \frac{1}{d_{ij}} \tag{4}$$

N is the total number of nodes in graph *G*. $\sum_{j \neq i} \frac{1}{d_{ij}}$ refers to the distance from node *i* to all other nodes.

**Cultural level and vitality level evaluation.** To obtain the cultural level of each grid, we first use the spatial join tool in GIS, and then we count the number of historical place names and historical buildings within each grid cell. Additionally, we determine whether the grid cell is located within a historical cultural district, with a count of 1 if it is and a count of 0 if it is not. The total counts are then classified into low, medium, and high categories using

the natural breaks method as the cultural level of each grid. Similarly, to evaluate the vitality level of each grid, the spatial join tool is used to calculate the sum of the number of POIs and check-in occurrences for each grid cell. These totals are also classified into low, medium, and high categories representing the vitality level of the grid.

## Experimental evaluation findings and analysis

### Basic information about the knowledge graph

Our knowledge graph comprises 257,149 entities and 6,228,004 relationships. It contains 47,251 historical and cultural dimension entities and 232,562 contemporary dimension entities, of which 22,664 region entities are excluded. Excluding the BorderBy (178,762) and NearBy (1,836,211) relationships, which indicate spatial proximity within the grid, there are 2,060,014 historical dimension relationships and 6,137,916 contemporary dimension relationships.

The network metrics derived from the analysis of the graph network are presented in Table 4. The results indicate that M1 has the highest ratio of instance count to ontology count, indicating the highest knowledge density; the M3 network has an average of approximately 12 connections per node, whereas M1 has 5 connections and M2 has the fewest connections. In terms of closeness centrality, the nodes in the M3 and M1 networks are more tightly connected, whereas those in the M2 network are relatively sparse.

### Embedding model performance evaluation

The changes of metrics during 100 epochs are illustrated in Fig 6. The trend of training loss shows that M1 and M3 have the lowest training loss, while M2 has a higher training loss. However, in terms of MRR, M1 and M3 show similar values, with M3 slightly surpassing M1 in later stages and M2 recording the highest value, indicating that M2's predictive search accuracy is superior. The Hit@10 metric also reveals that M2 has the highest search accuracy, followed by M3, which slightly outperforms M1. Additionally, both M1 and M3 demonstrate higher accuracy in head entity searches compared with tail entity searches.

From the metrics on the test dataset (shown in Table 5), a similar trend to that of the training dataset is observed. All three models have high predictive accuracy. M2 exhibits the most favourable parameters, with M3 being slightly better than M1. Notably, for both M1 and M3, the prediction accuracy for the tail entities is marginally higher.

### Embedding model validity testing

This study developed prediction and similarity calculation functions based on the KGE model. The prediction function outputs the possible entity in a triple with a missing entity. The similarity computation feature enables the calculation of cosine similarity between two given entities. Therefore, two experiments were designed to evaluate them.

**Table 4. Graph network evaluation metrics.**

| Metrics | M1 | M2 | M3 |
|---|---|---|---|
| Avg. Density | 12,761.30 | 6,355.91 | 8,571.63 |
| Avg. Degree Centrality | 4.73 | 0.24 | 11.72 |
| Avg. Closeness Centrality | 0.17 | 0.18 | 0.05 |

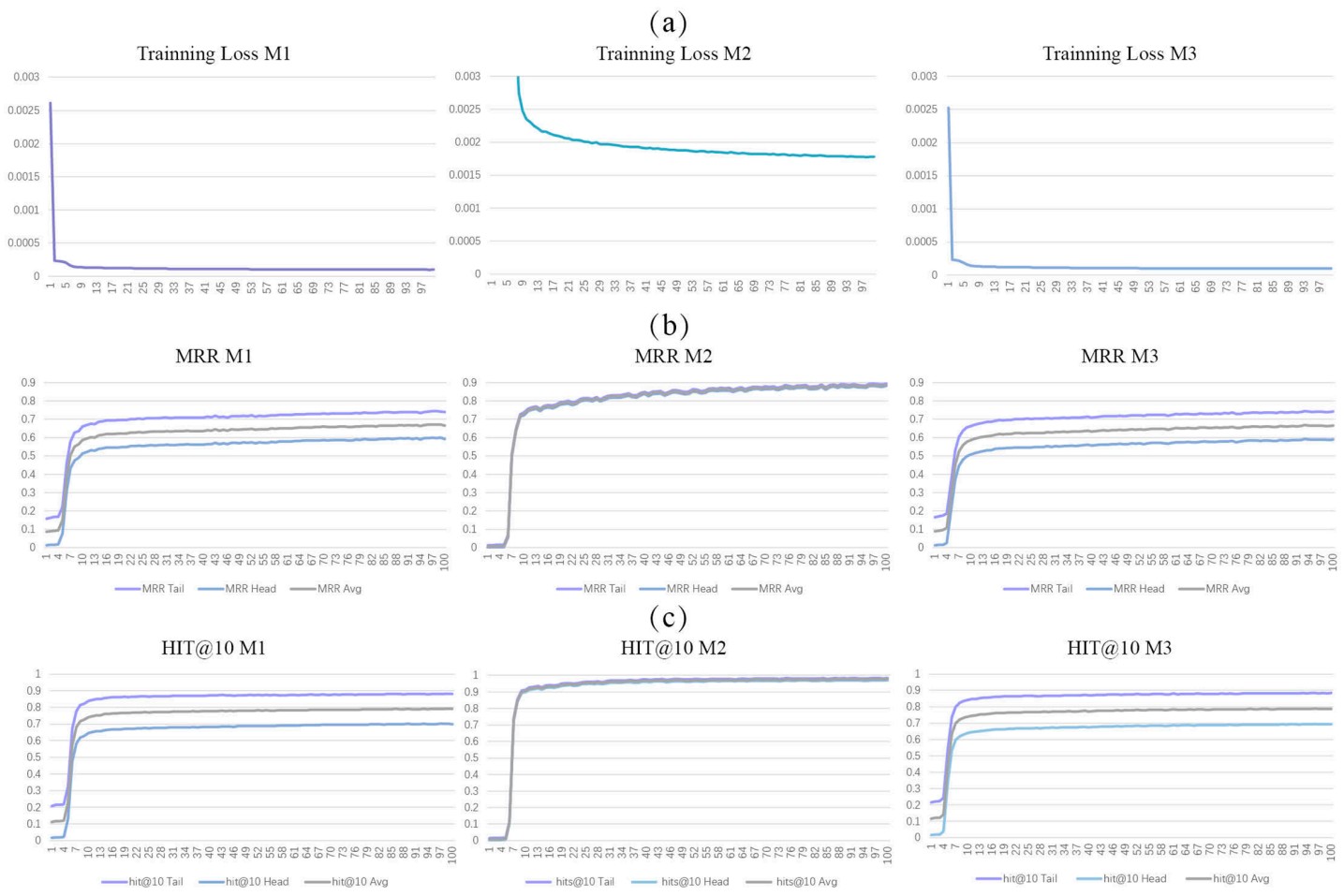

**Fig 6. The metrics of the three embedding models on the validation dataset across 100 epochs.** (a) Training Loss; (b) MRR; (c) Hit@10

**Table 5**. The metrics of the three embedding models on the test dataset.

|         | M1   |      |      | M2   |      |      | M3   |      |      |
|---------|------|------|------|------|------|------|------|------|------|
| Metrics | Head | Tail | Avg  | Head | Tail | Avg  | Head | Tail | Avg  |
| MRR     | 0.59 | 0.74 | 0.66 | 0.88 | 0.89 | 0.89 | 0.59 | 0.74 | 0.68 |
| hits@10 | 0.70 | 0.88 | 0.79 | 0.97 | 0.98 | 0.98 | 0.71 | 0.89 | 0.80 |
| hits@1  | 0.54 | 0.65 | 0.60 | 0.82 | 0.84 | 0.83 | 0.55 | 0.67 | 0.61 |

**Prediction validity testing.** We selected the cell with RegionID = 14931 as the subject. It is known that this grid has a *WorkedAt* relationship with 20 tail entities. The experiment requires the M3 model to output the top 20 and top 25 most likely tail entities that have a *WorkedAt* relationship with region 14931. The *WorkedAt* relationship denotes a home-work connection. For example, A *WorkedAt* B indicates that residents living in spatial unit A are employed at spatial unit B. The results are shown in Fig 7, and the complete results table is provided in S1 Table. The results indicate that when the model was tasked with generating predictions equal in number to the factual relationships, it produced 19 correct predictions, with only one erroneous output. When the model was required to predict more relationships

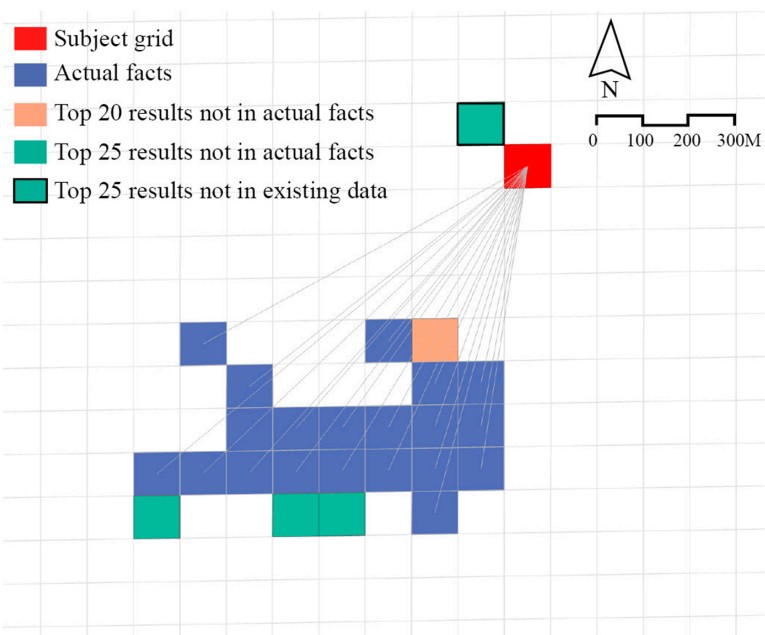

**Fig 7. Tail entities linked by the WorkedAt relationship with RegionID = 14931, facts and prediction results.**

than those present in the factual data, all 20 factual relationships were included in the predictions. The remaining five predictions, which did not correspond to factual data, were further examined for plausibility. The *WorkedAt* relationship requires more than five individuals to be working within a grid and living in another grid. By lowering this threshold, we found that four out of the five additional predictions were consistent with this relaxed criterion. The only unresolved prediction is the grid with RegionID = 14791, for which no supporting evidence could be found in the data. We contend that this result is well-reasoned. The grid with RegionID 14931 is situated in the central area of Haizhu District and represents an urban village surrounded by institutions such as research centres, residential areas, hospitals, and kindergartens. These serve as the primary employment hubs for residents within the urban village. The distribution of workplaces indicates that residents of the grid with RegionID 14931 tend to work in nearby locations. Adjacent to this grid is RegionID 14791, which contains small community-based shops, children's training centres, and kindergartens, all of which potentially provide employment opportunities for local residents. This highlights how the existence of urban villages significantly mitigates the spatial separation between living and working in cities, enabling residents to access income-generating employment opportunities within walking distance.

**Similarity calculation validity testing.** We selected the grids with RegionID 5338 and 4043 because of their high similarity in historical and cultural dimensions but distinct differences in contemporary dimension contexts. In the historical dimension, grid 5338 is situated in the Guangfu South Historical and Cultural District, which is known for its Time-Honoured Brand. Similarly, grid 4043 is in the Shangxiajiu Historical and Cultural District, also featuring established Time-Honoured Brand stores. Both of them associated with trade culture. In contrast, in the contemporary dimension, grid 5338 shows prominent residential characteristics, as it is situated outside modern commercial hubs and focuses on consumer shopping and life services. Conversely, grid 4043 is located in the Xiguan commercial area: it concentrates

on pharmaceutical and healthcare businesses and lacks a clear residential or work preference. We computed the cosine similarity for both grids using models M1, M2, and M3 shown in Table 6. The findings indicate a high similarity in historical contexts but a low similarity in contemporary dimensions. However, when both dimensions are considered together, the similarity is moderate, aligning with empirical observations. This phenomenon suggests that locations which appear entirely distinct today may have profound connections in historical and cultural dimensions.

## Application scenarios

This section presents four scenarios for the application of the model.

### Mining potentially similar places and entities

We select the spatial grids within the Guangfuzhong-Shangxiajiu district as the case study area. Our objective is to use the model to identify spatial candidates that possess cultural semantics closely aligned with this district.

The problem can be further defined as follows.

*Problem 1: Given a set of spatial grid entities set G located within the GuangfuZhong-Shangxiajiu district(hereafter Guang-Shang District), the task is to output another set of spatial grid entities G′ that are most similar to G considering only the historical dimension of semantics. Solution: Because Problem 1 focuses on the similarity of the historical and cultural dimension, input all spatial grids' RegionID within G into the M2 model. We use its Similarity function, output the top 20 entities that are most similar for each grid in G.*

The output entities are divided into two categories. The first category consists of spatial entities that are historically similar to the input locations, as shown in Fig 8. The second category includes non-spatial entities, as illustrated in Fig 9.

Fig 8 shows that similar spaces are distributed along the Pearl River, concentrated in the Xiguan Old District on the north bank and the Ho Nam Old District on the south. These areas include historical and cultural districts such as Guangfu South Road, Heping Middle Road, Yide Road-Sacred Heart Cathedral, the traditional central axis, Nanhua West Street, and Duobao Road-Baohua Road, as well as historic villages such as the Huangpu Ancient Port, the former Whampoa Military Academy site, Xiaozhou and Beiting historic villages.

Fig 9 presents non-spatial entities related to the case area, where the font size of each entity label reflects its similarity weight, such as Qiyi Road with its revolutionary culture links; well-known brands such as Yangcheng Hotel, Anya Pharmacy, Shenglongchang, Lianxianglou, and Datong Restaurant; and famous figures such as Wei Guoyao, Li Boming, and Mao Zedong who are associated with Red revolutionary culture.

### Destination recommendation by cross-retrieval

Through multi-step cross-retrieval using the prediction function, we can recommend possible destinations that meet users' various needs. The problem is defined as follows.

**Table 6**. The similarity between two grids.

| Input model | Similarity |
|---|---|
| M1 | 0.2282 |
| M2 | -0.0145 |
| M3 | 0.007 |

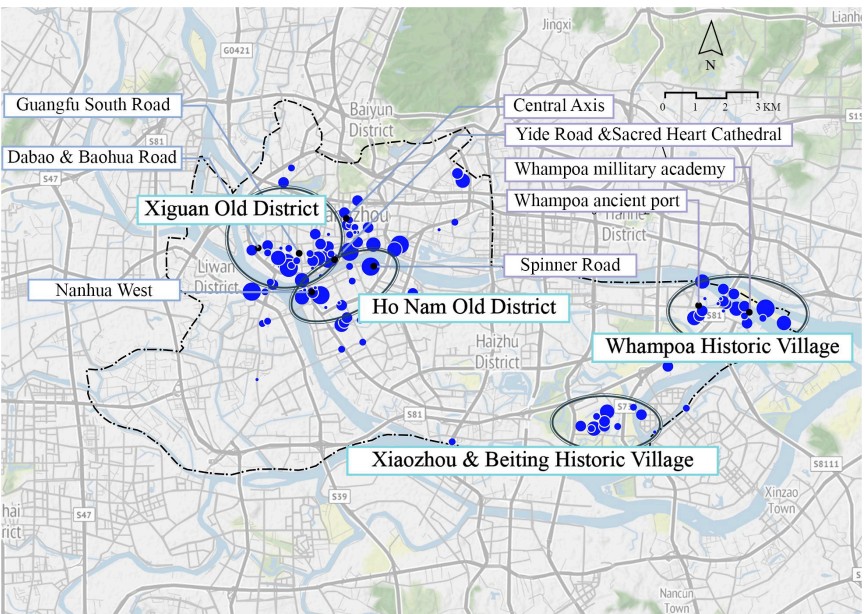

**Fig 8. Spatial entities exhibiting high cultural similarity to Guang-Shang District.** Image courtesy of the Earth Science and Remote Sensing Unit, NASA Johnson Space Center

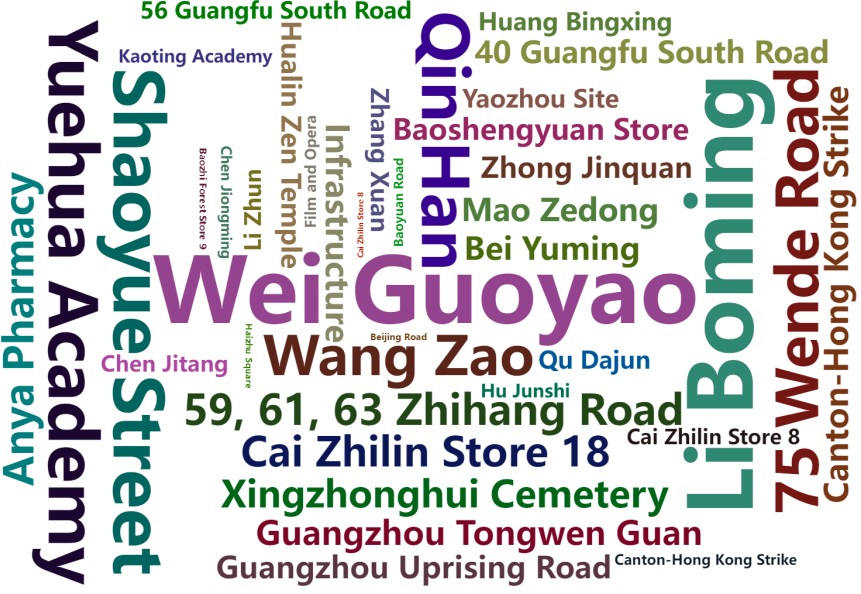

**Fig 9. Non-spatial entities exhibiting high cultural similarity to Guang-Shang District.**

*Problem 2: Given a set of user requirements R = {A ∧ B ∧ C}, A, B, and C are propositions translated from users' requirements. The objective is to find a subset of grids, s, from the set of all grids S that satisfies R.*

*Solution: Apply propositions A, B, and C to S successively, identify those satisfying all propositions to form s, and sort s based on importance.*

Assume that there is a resident with a known address who wishes to explore areas in Guangzhou rich in cultural heritage, preferably related to Red culture, and to visit bustling places near the commercial area. Thus, recommended destinations should meet these rules:

- Related to Red culture on cultural semantics
- Near business areas
- High vitality and cultural atmosphere
- Infrequently visited by locals who live near the requester

The retrieval steps are shown in Fig 10. First, we locate the user's request position, and then use the M3 model to filter all labels related to Red culture to generate the candidate set $s1$. Use *HasMainFunction* and *ProvideService* relationship prediction to filter those closely combined with commerce to generate $s2$. Use *ODFlow* relationship prediction to filter locations with a high probability of being visited by local residents to create $s3$. To find places less likely to be visited by local residents, perform a complement operation to obtain a complement set $s3'$. Finally, solve the intersection of $s1$, $s2$, and $s3'$ and sort based on their historical cultural atmosphere values to obtain the final recommended locations, as shown in Table 7.

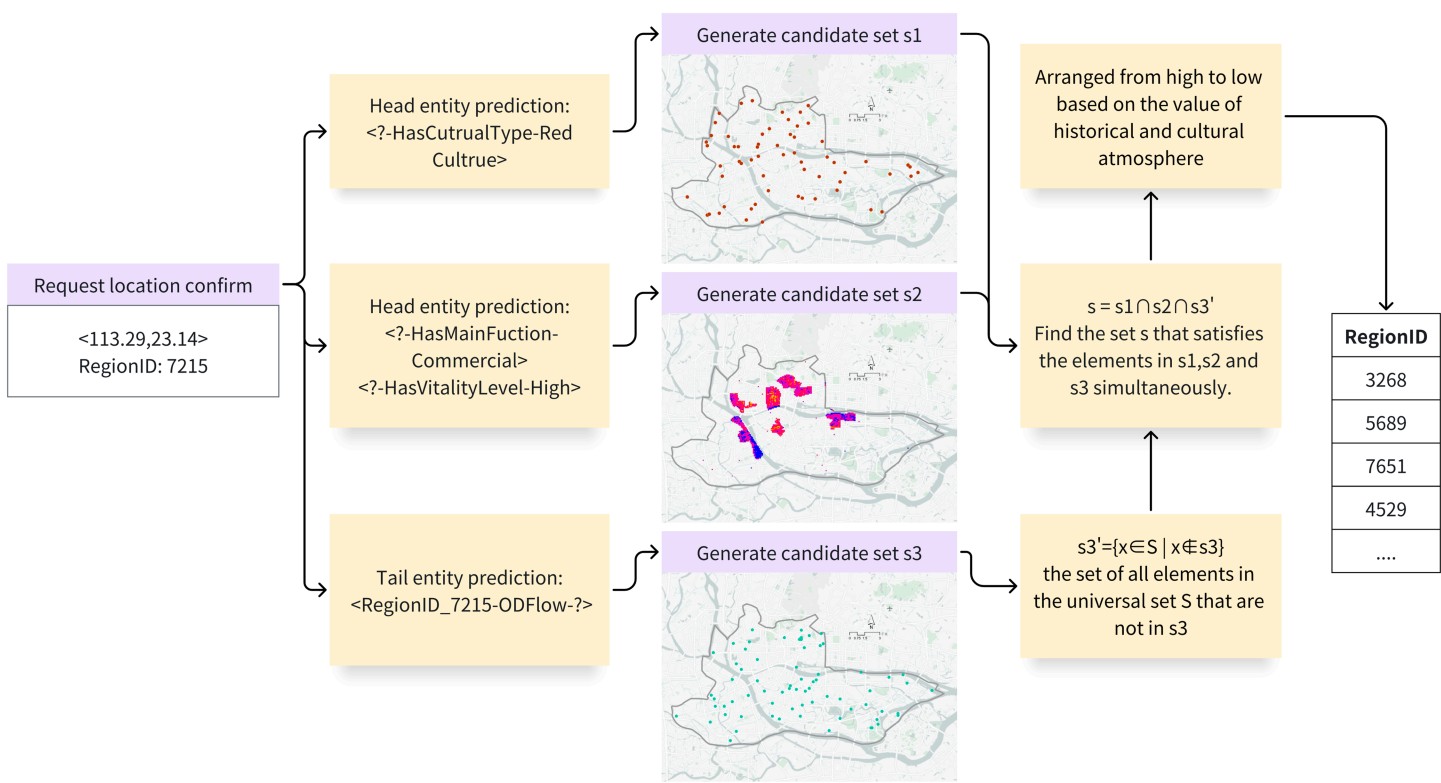

**Fig 10. Multi-step retrieval.**

**Table 7. Recommended results.**

| RegionID | Historical and cultural atmosphere | Related Red cultural heritage sites | Related business area |
|---|---|---|---|
| 3268 | High | Chen Tiejun and Zhou Wenyong's revolutionary base | Da XiGuan (Shang Xia Jiu - Yong Qing Fang) Business Area |
| 4529 | High | The former residence of Yang Pa'an | Beijing Road - Haizhu Square Business Area |
| 7651 | High | The Whampoa Military Academy Alumni Association's former site | Beijing Road - Haizhu Square Business Area |
| ... | | | |

## Time-Honored Brand site selection

A Time-Honoured Brand denotes an established brand with a long-standing legacy, typically spanning at least five decades, embodying distinctive cultural, artisanal, or commercial traditions. It preserves unique artisanal techniques, operational philosophies, or service characteristics rooted in cultural heritage, symbolising traditional craftsmanship and cultural memory. Time-Honoured Brand stores generally tend to be located in old districts that have a long history and thriving trade. However, as times change, these stores need to find suitable locations within new urban spatial structures to maintain their brand value. With an understanding of the spatial characteristics of the current distribution of time-honoured stores, the model can output more potential location choices.

*Problem 3: Given the spatial distribution grid set S of Time-Honored Brand stores, output a set N as new location choices for Time-Honored Brand stores.*

*Solution: Use the M3 model, which integrates historical and contemporary semantics, to perform similarity retrieval on all elements in S to form the new set N.*

The current and predicted locations are shown in Figure 11. The predictions focus on old Guangzhou's Xiguan, Ho Nam, and Dongshan areas, with increased numbers in Ho Nam and

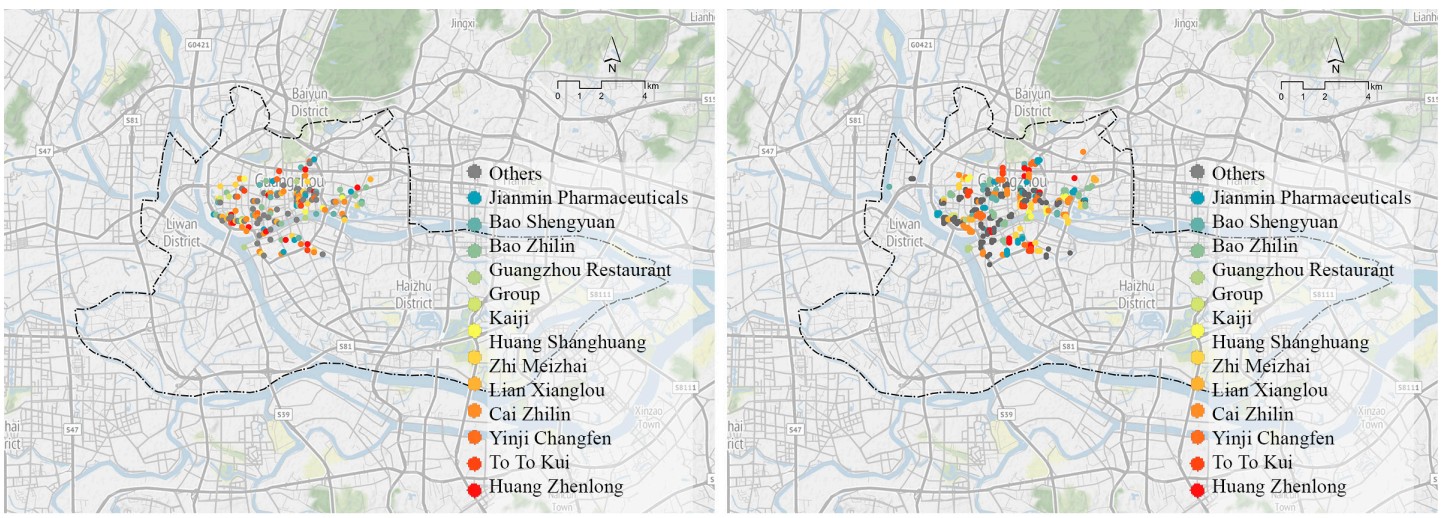

**Fig 11. Time-Honoured Brand current location and predicted site selection output results.** (a) Current location; (b) Predicted site selection. Image courtesy of the Earth Science and Remote Sensing Unit, NASA Johnson Space Center

Dongshan, which attract more customers from Tianhe (new urban central axis) and Haizhu Districts.

## Inferring a portrait of visitors

Additionally, this model can help managers of historic and cultural blocks to understand the demographic characteristics of their visitors, allowing them to adjust their promotional strategies to align with visitors' preferences. This scenario uses Guang-shang District as an example.

*Problem 4: Given that the grid set S contains all grids within Guang-shang District, infer the set F that involves the subsets F1, F2, F3...Fi of characteristics of users visiting S, where i represents the $i_{th}$ characteristic.*

*Solution: First, infer the origin of the visitor foot traffic to S. Then, based on the characteristics of the residential population within the spatial grid of the origin, it is feasible to conduct demographic profiling.*

The process and results of the demographic profiling are shown in Fig 12.

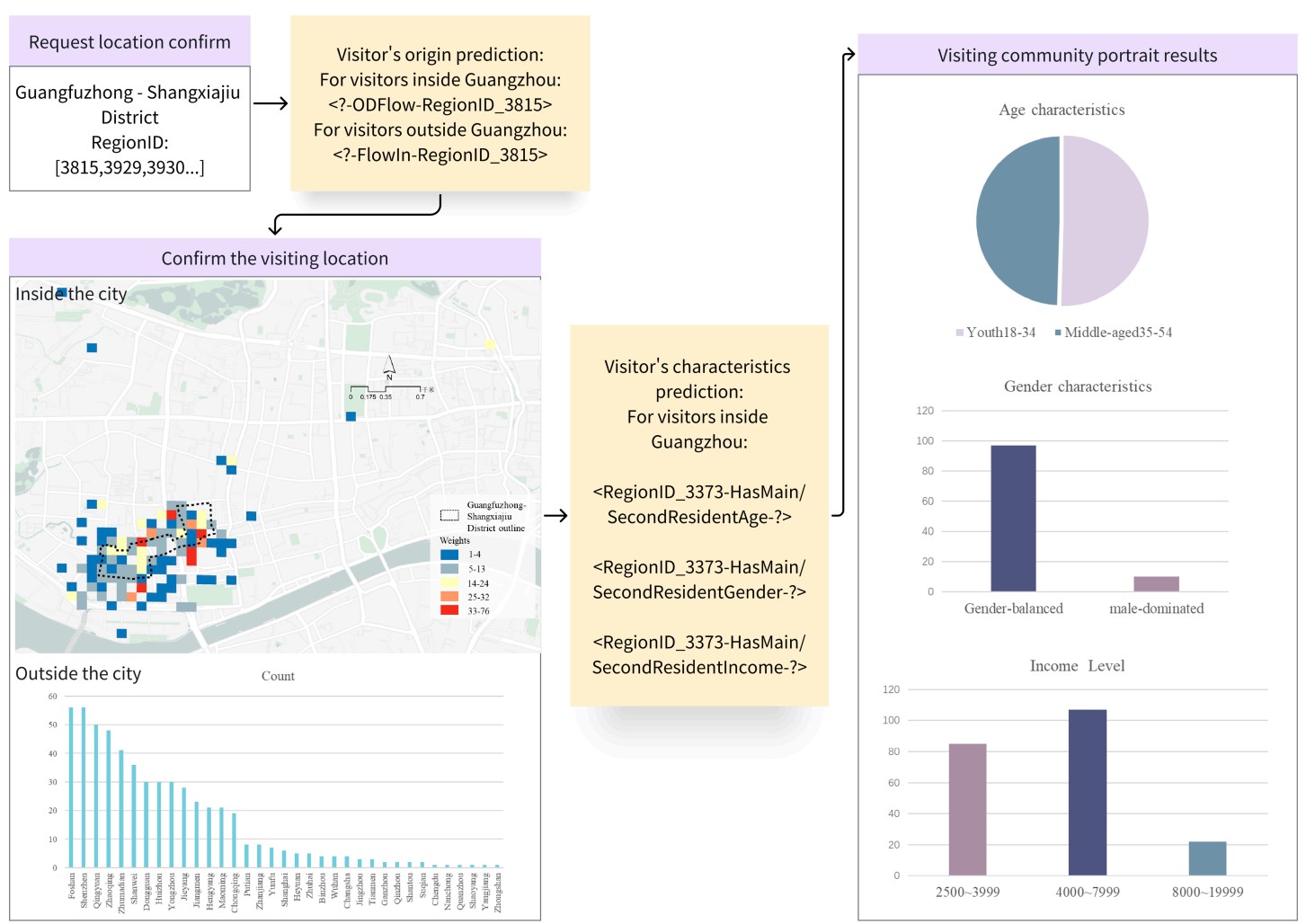

**Fig 12. Process of inferring a portrait of visitors.**

## Discussion of the effectiveness and practicability of the framework

In summary, using Guangzhou's ancient city district as a case study, this research demonstrates a framework for urban knowledge modelling (construction, embedding, and application) and empirically validates its operationally feasibility. The research hypothesis presented in the *Introduction* suggests the need for a holistic infrastructure to bridge historical-cultural and contemporary place semantics. The urban knowledge graph proposed in this study provides a solution to serve as such an infrastructure. It expands various historical semantics that contains historical figures, events, and related locations. As indicated by the quantitative analysis of the knowledge graph, it has formed a large network with numerous interconnections, approaching a fully connected network. Moreover, the proposed schema of the knowledge graph is expandable, allowing for new data sources to integrate additional types of entities by constructing new relationships on the existing base. This knowledge model not only aligns with the Historic Urban Landscape (HUL) approach's emphasis on the layered dynamism and richness of cultural heritage, but also embodies the principles of Digital Heritage by enabling continuous public engagement with evolving heritage narratives.

Regarding the second hypothesis mentioned in Section *Introduction*, to enable the application of the knowledge graph, we first performed graph embedding. The embedding supports both dual-dimensional and single-dimensional configurations, providing flexibility to adapt to specific scenario requirements. The embedded model offers functionalities such as prediction and similarity computation, with its reliability validated through evaluation experiments. Subsequently, we proposed four application pathways for the model in different scenarios, highlighting their potential value as follows:

- Reasoning about places with similar cultural semantics can uncover potential connections between locations, showcasing cultural links to the public or related scholars. For governments and cultural heritage protection agencies, it can provide scientific evidence for the systematic protection of cultural heritage.
- Developing multiple cross-references to recommend destinations themed around urban historical culture based on various user demands. Through recommendations and result sorting, personalised destination recommendations can be provided for individuals, increasing their exposure to related historical and cultural sites. Relevant departments can also identify sites or routes with potential for tourism development.
- Assisting Time-Honoured Brand stores in choosing locations that adapt to today's urban structures. These stores are living fossils of urban history; adapting to new urban structures and expanding their business can allow more visitors to enjoy their services and culture, and they can thus provide tourists with more resources.
- Inferring visitor profiles for scenic areas by analysing the demographics of visitors, enabling managers to accurately understand user behaviours and preferences to formulate appropriate marketing strategies. By extending this workflow to generate visitor profiles for additional historic districts or locations, comparative analysis of these profiles would enable relevant institutions to: (1) identify which historical and cultural semantics demonstrate stronger visitor attraction and their corresponding target demographics, (2) determine which cultural narratives resonate most with the public versus those receiving limited engagement, and (3) investigate the underlying causes of low visitation - whether due to inadequate promotion or other systemic factors requiring intervention.

In section *Application Scenarios*, the model's outputs demonstrate both rationality and creativity, uncovering implicit relationships that are not readily discernible through human

knowledge. Especially in the case of mining potentially similar places and entities, the output results exhibit cultural and temporal similarities with the Guang-Shang District. As the background, the Guang-Shang District is located within the Xiguan Old District. Guangfu Zhong is known for its Red culture, which primarily focused on journalism and played a significant role in spreading ideas during the Chinese People's War of Resistance. It also features the distinctive Xiguan Great Houses, renowned Time-Honoured Brand stores, and a historical textile industry. Shangxiajiu is marked by Cantonese opera, historic cinemas, arcade buildings, and a vibrant commercial history.

By identifying the above background, we can compare the output results with the Guang-Shang District itself. The western Duobao Road-Baohua Road and the southeastern Guangfu Nan areas are also in the Xiguan Old District. They have been greatly influenced by the trade dominance of commercial activities so that retains much of its urban texture, historical buildings(which contain arcade buildings), and renowned Time-Honoured Brand stores. The Yide Road-Sacred Heart Cathedral is characterised by arcade streets and religious culture and is closely tied to coastal commercial activities. Points within the central axis of the historical cultural district possess attributes of Red culture sites, arcade architecture, and commercial origins. In more distant areas, locations with similar cultural semantics were found. Points near Nanhua West Street in Ho Nam District used to be a hub for merchants and foreign traders, linking them to Guang-Shang District's commercial culture. Locations on Textile Road in Ho Nam District, surrounded by textile factories, resemble Guangfu Zhong's historical industries. The Whampoa Ancient Port is a crucial node for foreign trading vessels entering Guangzhou. It is strongly associated with commercial culture, and Whampoa Military Academy relates to the Anti-Japanese War, related to the red culture. Besides, the two historical villages are culturally unique in their architectural styles, clan culture, and folk beliefs. In addition, much of the historical heritage of Guang-Shang District dates back to the Qing dynasty and the Republic of China. Many of the other output locations also emerged during this era, reflecting the development of trade markets, businessmen-driven building construction, the formation of arcade streets, and the birth of revolutionary sites.

The three models (M1, M2 and M3) can be flexibly applied based on application scenarios' needs. If only the historical-dimension semantics of place are required, the model trained on the historical subgraph (e.g., Scenario 1) is preferable due to its superior historical relationship learning. For contemporary-dimension insights, M1 or M3 (e.g., Scenarios 2 and 4) are more suitable. When both historical and contemporary dimensions must be integrated, M3 should be used. However, since historical data demand more manual processing while contemporary data benefit from geospatial big data, the historical dataset is significantly smaller. This imbalance causes M3 to bias toward contemporary patterns during training, reducing its accuracy in historical relationship prediction. This is a limitation to address in future work.

## Conclusion and future work

To address the challenges associated with local historical-cultural semantics, align with UNESCO's policy initiatives, and bridge the existing research gap in integrating contemporary and historical-cultural semantics, this study proposes a methodological framework for developing an urban knowledge graph that synthesises historical-cultural and contemporary place semantics. This urban knowledge graph is enabled as foundational digital infrastructure for urban cultural preservation. After embedding, it demonstrates multi-scenario applicability in bridging cultural heritage with modern urban practices. The results indicate that the KGE model has accurate predictive capabilities; moreover, it can uncover hidden links that go unnoticed by humans. The inherent characteristics of knowledge graphs grant the framework

and model flexibility and scalability, allowing adaptation to various data types and focusing on user-relevant dimensions in different applications. Therefore, this study addresses a critical limitation in existing research—the isolated analysis of either historical or contemporary place semantics—by proposing an integrative framework for bidirectional semantic alignment. The framework demonstrates significant applied potential in cultural heritage protection, education, public engagement, cultural tourism, and real-time information services.

However, the work has some limitations:

- Historical data required additional manual processing, resulting in a significantly smaller dataset compared to contemporary data. In this study, separate models were trained for historical and contemporary dimensional subgraphs. The appropriate model can be selected based on the task type (e.g., Model M2 for tasks with a historical focus). In this study, models were separately trained for historical and contemporary dimensional data. The choice of the model depends on the semantic dimension emphasised by the predictive task (e.g., Model M2 should be employed for tasks focusing on the historical dimension). However, the jointly trained M3 model exhibited a critical bias. It may predominantly learn patterns from contemporary data, leading to inaccurate predictions for historical triples. To address this imbalance, future studies should recalibrate the loss function by assigning higher weights to historical dimension prediction tasks.
- This study employs 100×100m spatial grids as fundamental spatial units for data aggregation, a widely adopted approach in geographical research practices. Patches with significant spatial heterogeneity are grouped into the same grid cells, which may result in a deviation of the grid's aggregated attributes from the attribute values observed at smaller spatial scales. This scale effect primarily impacts contemporary dimension data aggregation. For historical dimension data, this study establishes one-to-multiple correspondence relationships for each grid unit to comprehensively capture all cultural semantics present within the grid, as the historical-cultural data is discontinuous.
- Furthermore, this study proposes a comprehensive methodological framework, with its efficacy empirically validated through a case study of Guangzhou. However, the KGE model trained on Guangzhou data demonstrates limited transferability to other urban contexts. This constraint stems from fundamental divergences in cities' historical development trajectories and contemporary urban morphologies.

## Supporting information

**S1 Table. Results of WorkedAt relationship prediction.**
(XLSX)

## Acknowledgments

Thanks to student research assistants Ms. Cheng Ying, Ms. Ni Liqiao, Mr. Sun Xuan and Ms. Xue Hanyu for their contributions on data collection and processing. Guangdong Architectural Design and Research Institute Group Co., Ltd. provided data supports. Professional English language editing support provided by AsiaEdit (asiaedit.com). We are deeply saddened by the passing of our co-author, Cuicui Xu, who contributed significantly to this work but unable to witness its publication. We respectfully dedicate this paper to her memory.

## Author contributions

**Conceptualization:** Chaoqun Wang, Cuicui Xu, Weijiang Pan.

**Data curation:** Chaoqun Wang, Cuicui Xu, Yinglin Wang, Xin Xu.

**Formal analysis:** Chaoqun Wang, Cuicui Xu, Yinglin Wang.

**Funding acquisition:** Jie He.

**Investigation:** Chaoqun Wang, Cuicui Xu.

**Methodology:** Chaoqun Wang.

**Project administration:** Jie He, Weijiang Pan.

**Resources:** Chaoqun Wang, Xin Xu.

**Software:** Chaoqun Wang, Yinglin Wang.

**Supervision:** Jie He, Weijiang Pan.

**Validation:** Chaoqun Wang.

**Visualization:** Chaoqun Wang, Cuicui Xu, Yinglin Wang.

**Writing – original draft:** Chaoqun Wang.

**Writing – review & editing:** Jie He.

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
