## [Decision Letter · Decision Letter 0]

30 May 2025

PONE-D-25-15113Construction and Application of a Novel Urban Knowledge Model with Extended Historical and Cultural SemanticsPLOS ONE

Dear Dr. He,

Thank you for submitting your manuscript to PLOS ONE. After careful consideration, we feel that it has merit but does not fully meet PLOS ONE’s publication criteria as it currently stands. Therefore, we invite you to submit a revised version of the manuscript that addresses the points raised during the review process.

We look forward to receiving your revised manuscript.

Kind regards,

Weicong Li, P.hD

Academic Editor

PLOS ONE

 [This work was supported by the Research Initiative Fund for Newly Introduced Talents of Harbin Institute of Technology, Shenzhen �#ZX20230488�]. 

4. In the online submission form, you indicated that [The data is available on request from the author].

5. We notice that your supplementary table is included in the manuscript file. Please remove them and upload them with the file type 'Supporting Information'. Please ensure that each Supporting Information file has a legend listed in the manuscript after the references list.

Additional Editor Comments:

Please find my comment in reviewer 7. The manuscript has research value, but still needs to address the following major issues, as well as respond to and adjust other reviewers' comments.

Reviewers' comments:

Reviewer's Responses to Questions

**Comments to the Author**

1. Is the manuscript technically sound, and do the data support the conclusions?

Reviewer #1: No

Reviewer #2: Yes

Reviewer #3: Partly

Reviewer #4: Yes

Reviewer #5: Yes

Reviewer #6: Yes

Reviewer #7: Partly

2. Has the statistical analysis been performed appropriately and rigorously? 

Reviewer #1: No

Reviewer #2: Yes

Reviewer #3: N/A

Reviewer #4: Yes

Reviewer #5: Yes

Reviewer #6: Yes

Reviewer #7: N/A

3. Have the authors made all data underlying the findings in their manuscript fully available?

Reviewer #1: No

Reviewer #2: Yes

Reviewer #3: No

Reviewer #4: No

Reviewer #5: No

Reviewer #6: Yes

Reviewer #7: Yes

4. Is the manuscript presented in an intelligible fashion and written in standard English?

Reviewer #1: Yes

Reviewer #2: Yes

Reviewer #3: No

Reviewer #4: Yes

Reviewer #5: Yes

Reviewer #6: Yes

Reviewer #7: Yes

5. Review Comments to the Author

Reviewer #1: In the 5th line, a direct transition to machine learning has been made, artificial intelligence should be mentioned before.

Contributions are not explained concretely and clearly

Literature studies on the subject are missing

Explanations of the figures are missing

Where is the dataset? Was it shared as an open link or are there any benchmark datasets on this subject?

Why were graph-based structures not preferred? Like graph neural network

How was the batch size or optimization algorithm selection made in model training? Was early stopping done? In Figure 6 (Model overfitting is visible).

How were the evaluation metrics selected, are there any other metrics?

There is no transition to Subject integrity in the entire article

No references Neo4j

Figure 9 is a word cloud shape?

There is no connection between quantitative and qualitative results.

Send feedback

Side panels

History

Saved

Reviewer #2: The manuscript titled "Construction and Application of a Novel Urban Knowledge Model with Extended Historical and Cultural Semantics" presents an innovative and methodologically ambitious framework for integrating historical and contemporary semantics in urban spatial analysis. It introduces a multilayered urban knowledge graph (UrbanKG), applying knowledge graph embedding technologies to bridge fragmented historical data with present-day urban datasets. The study offers promising applications for heritage management, urban planning, and cultural tourism, using the historic city of Guangzhou as a compelling case study.

The research is grounded in the Historic Urban Landscape (HUL) framework and seeks to operationalize it through the design of a semantic knowledge infrastructure. This is a highly relevant and timely contribution, especially in light of increasing interest in digital heritage, the role of AI in planning, and the democratization of historical knowledge through computational platforms. The construction of UrbanKG reflects a sophisticated integration of unstructured archival data, GIS-based spatial analysis, and contemporary geo-big data.

The strength of this manuscript lies in its comprehensive methodology. It combines the digitization of cultural heritage assets with data-driven urban informatics, layered through a knowledge graph that supports entity prediction, semantic similarity, and retrieval. The schema is well-conceived, and the embedding models are rigorously evaluated using accepted metrics such as MRR and HITS@10. The inclusion of practical application scenarios — including personalized heritage recommendations, predictive siting of Time-Honored Brands, and visitor profiling — demonstrates the potential of this tool for both scientific and operational purposes.

However, several dimensions of the paper require improvement to enhance its scientific depth, theoretical integration, and clarity of exposition. These are outlined below as part of a structured set of recommendations.

1. Strengthen Theoretical Engagement and Conceptual Framing: While the manuscript effectively adopts the HUL framework and introduces knowledge graph technologies, it would benefit from deeper theoretical engagement with literature on digital heritage, place-based semantics, and spatial humanities. Concepts such as "cultural memory," "urban narrative," and "semantic place theory" could provide a richer interpretive lens and help the study speak more broadly to international academic debates.

2. Clarify Research Objectives and Contributions in the Introduction: The introduction presents a compelling case for the relevance of integrating historical and contemporary semantics, but the structure would be improved by clearly enumerating the study's research questions and hypotheses. The objectives are somewhat embedded in the narrative; making them more explicit would help orient the reader.

3. Enhance Methodological Transparency: The methodology is highly complex, yet several stages (e.g., the grounded theory tagging system for cultural types, the threshold selection for OD and workplace flows, and the justification for temporal segmentation) are described in a fragmented or insufficiently detailed manner. A tabulated summary of the workflow steps, supported by schematic diagrams, would improve replicability and transparency.

4. Refine Language and Presentation: The manuscript would benefit significantly from English language editing. Several sections are overly long, with inconsistent terminology and occasional grammatical errors. Improving the clarity and coherence of the writing will greatly enhance the manuscript’s readability and impact.

5. Critically Reflect on the Limitations and Ontological Biases: The paper acknowledges the imbalance between contemporary and historical data but does not sufficiently explore the implications. A more reflective discussion of biases introduced by the dominance of contemporary data sources, the manual processing of historical information, and the assumptions embedded in spatial units (e.g., 100m x 100m grid cells) would strengthen the epistemological rigor.

6. Expand the Discussion of Applicability and Transferability: The model is applied to Guangzhou, a rich and data-accessible case. It would be useful to address how UrbanKG might be adapted to other urban contexts, especially those with less structured data or more contested heritage landscapes. A comparative or scenario-based reflection could broaden the scope of the study’s contributions.

7. Deepen the Analytical Interpretation of Results: Much of the evaluation, particularly in Sections 3.1–3.3, focuses on performance metrics. While this is important, a stronger interpretive narrative is needed to connect these technical outcomes to substantive urban issues. For example, how might high semantic similarity between grids translate into planning insights, or how do the model's predictions align (or not) with existing policy frameworks?

Reviewer #3: This paper presents a knowledge graph-based framework integrating historical and contemporary semantics of urban spaces. Using data from Guangzhou, it merges structured and unstructured data sources—including Baidu APIs and archival documents—to construct an urban knowledge graph. The graph is embedded using ConvE, enabling similarity search, entity prediction, and cross-retrieval. Four application scenarios demonstrate the potential for cultural heritage engagement, including identifying culturally similar sites, recommending destinations, suggesting new locations for legacy businesses, and profiling visitor demographics. The work is conceptually aligned with UNESCO’s HUL framework and aims to support sustainable urban cultural development. While technically promising, the paper requires significant revisions to meet its conceptual and empirical claims.

Major Observations

• The paper introduces a strong technical framework integrating historical and contemporary urban semantics via a knowledge graph and embedding model.

• The two core objectives—building a systematic urban knowledge model and promoting public awareness—are only partially achieved. The first is well addressed; the second lacks implementation, validation, or user engagement.

• The research gap is real but overstated. The paper overlooks existing public-facing historical KGs like World Historical Gazetteer (WHG), Linked Places, and Recogito, which already support semantic historical exploration. A more nuanced comparison would better position the contribution.

• The four application scenarios are promising but remain simulated. No real-world deployment, user testing, or institutional collaboration is shown.

• Claims about planning support and cultural sustainability are made but not operationalized with metrics or institutional partnerships.

• Public engagement is claimed but not measured. There are no user-facing interfaces, behavioral data, or feedback mechanisms.

• Historical data imbalance is acknowledged but not addressed—triples are unweighted, and manual data curation is not algorithmically compensated.

• The model is trained and tested solely on Guangzhou. No replication or transferability to other cities is explored.

• The use of behavioral data (e.g., Weibo check-ins) is not accompanied by any discussion of ethics, anonymization, or review protocols.

• Data transparency is insufficient—no sample triples, pipeline diagrams, or data access options are provided, limiting reproducibility.

• The use of ConvE is not justified or compared with other models. No explanation is given for choosing it over alternatives like TransE, RotatE, or CompGCN, despite the spatiotemporal nature of the data.

• The terms “space” and “place” are used interchangeably without defining their conceptual distinction, which is important in urban semantics.

Minor Observations

• Line 186: “functions contains” → should be corrected to “functions that include”.

• Line 28: “aims to bridge” → should be “aim to bridge” (plural subject).

• Line 454: Duplicate word — “the the” should be corrected.

• Line 503: Clarify “limits regression tasks” → “limits support for regression tasks.”

• Line 422: Missing cross-reference — “Section ?” should be specified.

• Figure 2: Spelling error — “sematics” → should be “semantics.”

• Figure 5: Nearly identical to Figure 2 and possibly redundant; consider merging or removing.

• Figure 2: Mentions a “Memory Platform” that is never explained in the text—should be clarified or removed.

• Several phrases (e.g., “public-facing cultural semantics,” “cultural tagging system,” “WorkedAt”) use non-standard or unclear terminology and should be refined for clarity.

• The term “Time-Honored Brand” is central to one application scenario but is not defined; a brief explanation would aid understanding.

• The term “semanticization” should be replaced with the more conventional “semantic enrichment.”

• “Panoramic knowledge” is used metaphorically and repeatedly but is not a standard term. It should be replaced with “comprehensive” or “holistic,” or defined clearly if retained.

The paper presents a promising and timely contribution at the intersection of geospatial semantics, digital heritage, and urban computing. However, the conceptual framing is overstated in places, public impact is unvalidated, and key terms and components are either unclear or underexplained. Addressing these issues through clarification, justification, and more rigorous validation will significantly strengthen the manuscript.

Reviewer #4: Comments to the Author

Dear Authors,

The study titled “Construction and Application of a Novel Urban Knowledge Model with Extended Historical and Cultural Semantics” presents a knowledge-based model designed to bridge contemporary and historical semantics of urban spaces by leveraging geo-big data and digitized cultural heritage records. Focused on the ancient districts of Guangzhou, it showcases a methodology for integrating diverse semantic layers into a structured, query-able graph database. Prior works were thoroughly reviewed, and some important gaps in the literature have been clearly identified. Also, the manuscript is methodologically sound. However, I have some recommendation and comments for the authors.

- The methodological flowchart (Figure 2) needs to be revised. The text in the figure, especially on the top right, is not legible. I suggest the author(s) use a more legible font type and color.

- Can the author(s) provide citation for the grounded theory workflow used in developing a cultural tagging system. This will strengthen the credibility of the approach.

- I recommend that the author(s) should provide citation for the knowledge graph embedding (KGE) technology that was used in obtaining the vectorized representations of the triples. I believe many studies have used the same approach. Also, it would be great if the author(s) could justify the choice of the ConvE model for knowledge graph embedding over alternatives like TransE or DistMult.

- The author should double check the first part of the sentence in line 411-413 and make proper reference of the appropriate section. Also in line 422. The author must have omitted the Section number.

Reviewer #5: The paper "Construction and Application of a Novel Urban Knowledge Model with Extended Historical and Cultural Semantics" is technically sound and written in a very standard fashion, addressing a very interesting issue. The analysis has been performed rigorously. No comments to the authors.

Reviewer #6: Dear authors,

I have carefully read the manuscript entitled “Construction and Application of a Novel Urban Knowledge Model with Extended Historical and Cultural Semantics”. The document proposes a comprehensive and innovative methodology for the construction of an urban knowledge graph that fuses contemporary semantics derived from massive spatial data with historical semantics drawn from heritage sources. The approach is highly relevant in the context of historic urban landscape conservation and is aligned with the recommendations of UNESCO's HUL approach. The conceptual model, the architecture of the graph, and the practical applications described - such as the prediction of locations for heritage landmarks or the recommendation of cultural destinations - make a significant contribution to the crossover between computational urban science and the study of cultural heritage.

From a methodological perspective, the study relies on consolidated techniques such as knowledge graph embedding (KGE), visualization tools in Neo4j and robust semantic segmentation, both historical and spatial. Nevertheless, a more critical discussion of the accuracy and limitations of the manual annotation process of historical data would be desirable, as well as a more explicit quantitative assessment of the impact of integrating historical dimensions on the predictive performance of the combined model (M3). Although standard metrics such as MRR and Hits@n are mentioned, the manuscript could improve if a deeper discussion of model generalization outside of the Guangzhou case study was undertaken. Another minor issue is about Fig. 1, which must show the study area at national level, because the non-Chinese readers could not know where the study has been undertaken.

Finally, the article stands out for its practical relevance and scalability potential, but it lacks a greater problematization of the ethical, political or social implications of the public use of historical information in urban planning. A deeper reflection on the sustainability of the proposed model would also be pertinent, especially with respect to the continuous updating of the graph and the governance of the integrated data. Still, I consider that the paper presents a solid foundation, methodologically rigorous and with valuable applications, and with some minor revisions could be accepted for publication.

Kind regards

Reviewer #7: 1. It is suggested that the literature review could be expanded to also focus on how planners and policy makers are contributing to research in the subject area.

2. Adding an academic contribution map to section “Discussions” could be effective in revealing how research contributes to the field.

6. PLOS authors have the option to publish the peer review history of their article (what does this mean?). If published, this will include your full peer review and any attached files.

Reviewer #1: No

Reviewer #2: **Yes: **Pedro Chamusca

Reviewer #3: No

Reviewer #4: No

Reviewer #5: No

Reviewer #6: No

Reviewer #7: No

---

## [Author Response · Author response to Decision Letter 1]

24 Jun 2025

The following text may have formatting issues affecting readability. Could you please prioritize reviewing the 'Response to Reviewers.docx' file I have uploaded?

Dear Editor and Reviewers:

Thank you for your thorough review of our manuscript and for providing such valuable and constructive feedback. We sincerely appreciate the time and effort you dedicated to evaluating our work. Based on your insightful suggestions, we have carefully revised the manuscript to address all comments. Below is a summary of the key revisions made:

1. We have expanded the background discussion to provide a more rigorous foundation for the study’s motivation, clearly delineating the significance of our work.

2. The research questions, hypotheses, and contributions have been further clarified. The paper now explicitly presents a methodological framework spanning modeling to application, with empirical validation using a case study from Guangzhou.

3. The scope of the literature review has been broadened to include a more systematic synthesis of prior studies, with additional comparisons and critical analysis to better position our work within the existing scholarship.

4. The discussion now more tightly aligns with the research objectives, elaborating on how the study achieves its proposed goals while exploring its broader implications and potential applications.

5. The conclusion has been augmented with a deeper reflection on the study’s limitations, particularly regarding data imbalance, spatial unit selection, and model generalizability, providing avenues for future research.

We believe these revisions have significantly strengthened the manuscript and are grateful for the opportunity to improve it. Please do not hesitate to contact us if further clarifications or adjustments are needed.

Thank you again for your time and consideration.

Reviewer 1

1. In the 5th line, a direct transition to machine learning has been made, artificial intelligence should be mentioned before.

With regard to the comment on line 5 (assumed to refer to line 35 in the full manuscript, i.e., the fifth line of Introduction Section 2), we acknowledge that the original transition from urban historical semantics projects to machine learning applications in urban informatics appeared abrupt.

Both urban historical semantics projects and urban informatics research are underpinned by artificial intelligence technologies, the former aligns with symbolic AI approaches, while the latter employs connectionist AI techniques. We have incorporated this clarification in the revised manuscript, see line 37-51.

2. Contributions are not explained concretely and clearly

The previous version might have caused some misunderstanding regarding the description of this paper's contributions. In the Introduction section in this revised version, we have clarified the contributions as follows: Our contribution is the development of a comprehensive methodological framework that encompasses the construction of a knowledge graph integrating both contemporary and historical dimensions, the training of embedding models, and their application scenarios. The viability of this framework is empirically demonstrated through a case study conducted in Guangzhou. �see line 63-67

3. Literature studies on the subject are missing

We sincerely appreciate the reviewer’s insightful observation regarding the literature review on the subject. The reviewer rightly points out the need to contextualize our work within existing research on urban semantic extraction and structure both historical and contemporary.

However, we acknowledge that few studies explicitly bridge these two dimensions. This gap motivates our work’s core contribution which propose a framework to integrate historical-cultural and contemporary semantics into an urban knowledge graph.

Our framework is not developed in isolation. We integrated approaches from Reference[31],[33] and [92]. Reference [92] provided the technical framework and methodological reference for extracting contemporary urban semantics. Reference [31] and [33] inspired the extraction of historical-cultural semantics in this study.

In this revision, we have supplemented the literature review with additional fields to better compare and position the research presented in this paper. We have included open-source projects on historical place semantics, as well as attempts by governments and administrative bodies to use digital means to promote and showcase local cultural content. Please see "Extracting historical and cultural semantics in the digitization of cultural heritage" and "research gaps" section.

We hope this revision addresses the reviewer’s concern and more clearly establishes our work’s novelty.

4. Explanations of the figures are missing

We have carefully reviewed the figure captions and the main body of manuscript, revising some figure captions (e.g., Figures 1, 5, 6, 8, and 9) to more clearly describe the content of the figures. Additionally, for some figures (e.g., Figure 2), we have added further explanations in the main text to enhance the interpretation of their content.

5. Where is the dataset? Was it shared as an open link or are there any benchmark datasets on this subject?

Thank you for your reminder regarding the dataset availability. We have now uploaded our research dataset. It is accessible via https://github.com/DaqunW/UrbanKG_GZ_case

6. Why were graph-based structures not preferred? Like graph neural network

We sincerely appreciate the reviewer’s insightful question regarding our choice of knowledge graph embedding (KGE) methods. The reviewer rightly points out that graph neural networks (GNNs) are indeed well-suited for processing graph-structured data and could also be applied to KGE. However, after careful consideration, we opted for the ConvE model rather than GNN-based approaches for the following reasons:

1. Our study primarily relies on link prediction as a key function of knowledge graph embedding. While GNNs excel at generating node/entity embeddings, they do not inherently perform relation prediction—instead, they require additional decoder design to achieve this. In contrast, ConvE is specifically designed for triple-based link prediction, making it a more direct and efficient choice for our task.

2. Existing literature supports that ConvE outperforms GNN-based models in certain scenarios. Dettmers T et al (2018) demonstrated that ConvE achieves superior performance on benchmark datasets like FB15k-237, particularly in link prediction tasks, while maintaining higher parameter efficiency compared to GNNs.

3. While some GNN variants show advantages in tasks requiring joint modeling of nodes and relations (e.g., node classification or graph classification), ConvE remains highly competitive for simpler link prediction tasks, which aligns with our focus.

We acknowledge that our initial submission did not sufficiently justify the model selection. In response, we have expanded the discussion in Section Knowledge graph embedding to explicitly compare ConvE with other alternatives. We hope this clarification addresses the reviewer’s concern.

7. How was the batch size or optimization algorithm selection made in model training? Was early stopping done? In Figure 6 (Model overfitting is visible).

We sincerely appreciate the reviewer's valuable questions regarding the model training setup and potential overfitting. In this study, we empirically set the batch size to 100. While overfitting generally harms generalization, its effect is mitigated in our specific context for three reasons:

1. This study prioritizes a novel pipeline (from data processing → KG embedding → downstream applications) rather than building a universally generalizable model. The Guangzhou case study serves as a proof of concept to demonstrate the framework’s feasibility and utility. This study does not propose a generalized model applicable to any city.

2. For our application scenarios, higher prediction accuracy is prioritized over generalization. Overfitting here helps the model capture Guangzhou-specific patterns, which aligns with our goal of maximizing accuracy for this city’s unique data.

3. Urban knowledge graphs are highly city-dependent due to geographical, historical, and cultural variations. Thus, even a "perfectly" generalized model would require training for new cities. Each city's model should train independently to ensure accurate results.

We have included an analysis of the model’s generalization limitations in the Discussion section (see line 611-616), clarifying its scope and potential improvements for broader applications.

8. How were the evaluation metrics selected, are there any other metrics?

During experiments, we measured multiple metrics including Training Loss, MR (Mean Rank), MRR (Mean Reciprocal Rank), and Hits@1/10. These indices are commonly used in KGE tasks. Training Loss is a fundamental indicator of model convergence. MRR + Hits@n is a widely adopted combination that balances accuracy (via MRR) and robustness (via Hits@n). We ultimately retained Training Loss, MRR, and Hits@10 in the manuscript. Because MR and MRR, as well as Hits@1 and Hits@10, exhibited highly correlated trends in our results. Hits@10 provides a more lenient and informative measure of model performance compared to Hits@1, while MRR captures the overall ranking quality.

9. There is no transition to Subject integrity in the entire article

We sincerely appreciate the reviewer's valuable feedback regarding the paper's transition to its core themes. In this revised version, we have thoroughly restructured both the 'Discussion' and 'Conclusion' sections to better articulate and reinforce our research focus.

In the discussion section, we systematically address the two research hypotheses proposed in the Introduction. First, we demonstrate the effectiveness of constructing an urban knowledge graph that integrates both historical and contemporary semantics as a holistic digital infrastructure infrastructure. Second, we validate the proposed application scenarios' efficacy and potential value, supported by detailed interpretations of the model's output results.

The conclusion section has been strengthened to Reaffirm the study's contributions in:

1. Addressing the challenges of historical-cultural place semantics

2. Explicitly connect our work to UNESCO's policy initiatives

3. Clearly position the significance of our methodological approach within the research gap

We believe these revisions have substantially improved the paper's focus and clarity.

10. No references Neo4j

We sincerely appreciate the reviewer's valuable observation. In response, we have supplemented section 'Graph database evaluation and visualization' with a reference to a study that employs Neo4j for processing large-scale SNS data while evaluating its performance. This addition serves to demonstrate Neo4j's general applicability in handling relational data. See line 314.

11. Figure 9 is a word cloud shape?

Yes. In Section 'Mining potentially similar places and entities', we let the model to output all entities that are semantically proximate to the spatial grid cells within Guang-Shang District. The model calculates pairwise similarity between entities and ranks the results in descending order. For clearer visualization, we presented spatial entities in Figure 8, and displayed non-spatial entities in Figure 9. The font size of each term in Figure 9 corresponds to their similarity score.

12. There is no connection between quantitative and qualitative results.

The qualitative findings in this study are derived from quantitative computations, including link prediction and entity similarity calculation. Specifically:

1. In the first application scenario, both the spatial entities (Figure 8) and non-spatial entities (Figure 9) were obtained through entity similarity calculations performed on spatial grid units within the Guang-Shang District.

2. The destination recommendation results in the second application scenario were generated through multiple head/tail entity prediction operations.

3. The third application scenario follows a similar approach to the first one, where recommended locations for new branches of time-honored brands were determined by calculating entity similarity based on existing time-honored brand locations.

4. The fourth scenario involved head/tail entity predictions for specified relationship types (population movement), followed by aggregated statistical analysis to generate visitor demographic profiles.

Reviewer 2

1. Strengthen Theoretical Engagement and Conceptual Framing: While the manuscript effectively adopts the HUL framework and introduces knowledge graph technologies, it would benefit from deeper theoretical engagement with literature on digital heritage, place-based semantics, and spatial humanities. Concepts such as "cultural memory," "urban narrative," and "semantic place theory" could provide a richer interpretive lens and help the study speak more broadly to international academic debates.

We sincerely appreciate the reviewer's valuable suggestions regarding theoretical foundations. In response, we have substantially restructured the “Introduction” section (see line 1-36) to incorporate deeper theoretical engagement. The revised introduction now establishes its theoretical foundation through Semantic Place Theory, systematically explicating the concept of place semantics and its significance. In this version, we position historical and cultural semantics as a critical dimension of place semantics, currently facing dual challenges of gradual dissolution and public oblivion. Beyond the Historic Urban Landscape (HUL) framework, we have incorporated UNESCO's digital heritage policy documents to strengthen policy-oriented motivation. We believe this restructuring provides stronger theoretical anchoring.

2. Clarify Research Objectives and Contributions in the Introduction: The introduction presents a compelling case for the relevance of integrating historical and contemporary semantics, but the structure would be improved by clearly enumerating the study's research questions and hypotheses. The objectives are somewhat embedded in the narrative; making them more explicit would help orient the reader.

We are most grateful to the reviewer for raising this pertinent observation. Our initial submission didn't adequately situate the research problem, objectives, and contributions, which regrettably led to understandings.

In this revised version, we have thoroughly restructured the 'Introduction' section to clearly set out the research question, hypotheses and objectives, see line 52-67.

Our research question is:

【How can digital technologies construct an effective bridge between historical-cultural and contemporary place semantics】

Our research hypotheses are:

1. Resolving this question requires a holistic digital infrastructure capable of integrating the comprehensive semantics of urban places, encompassing both contemporary and historical-cultural dimensions.

2. Furthermore, practical implementation pathways must be outlined to maximise the potential of such an infrastructure.

Our contribution is:

We develop a comprehensive methodological framework that encompasses the construction of a knowledge graph integrating both contemporary and historical dimensions, and their application scenarios. The viability of this framework is empirically demonstrated through a case study conducted in Guangzhou.

3. Enhance Methodological Transparency: The methodology is highly complex, yet several stages (e.g., the grounded theory tagging system for cultural types, the threshold selection for OD and workplace flows, and the justification for temporal segmentation) are described in a fragmented or insufficiently detailed manner. A tabulated summary of the workflow steps, supported by schematic diagrams, would improve replicability and transparency.

We sincerely appreciate the reviewer's invaluable suggestion. In the original version, section 'Methodology' proceeded directly to explain individual steps without providing an overview of the complete workflow. To address this, we have implemented the following improvements

---

## [Decision Letter · Decision Letter 1]

13 Aug 2025

Construction and Application of a Novel Urban Knowledge Model with Extended Historical and Cultural Semantics

PONE-D-25-15113R1

Dear Dr. He,

We’re pleased to inform you that your manuscript has been judged scientifically suitable for publication and will be formally accepted for publication once it meets all outstanding technical requirements.

Kind regards,

Weicong Li, P.hD

Academic Editor

PLOS ONE

Additional Editor Comments (optional):

-

Reviewers' comments:

Reviewer's Responses to Questions

**Comments to the Author**

1. If the authors have adequately addressed your comments raised in a previous round of review and you feel that this manuscript is now acceptable for publication, you may indicate that here to bypass the “Comments to the Author” section, enter your conflict of interest statement in the “Confidential to Editor” section, and submit your "Accept" recommendation.

Reviewer #4: All comments have been addressed

Reviewer #6: All comments have been addressed

2. Is the manuscript technically sound, and do the data support the conclusions?

Reviewer #4: Yes

Reviewer #6: Yes

3. Has the statistical analysis been performed appropriately and rigorously? 

Reviewer #4: Yes

Reviewer #6: Yes

4. Have the authors made all data underlying the findings in their manuscript fully available?

Reviewer #4: Yes

Reviewer #6: Yes

5. Is the manuscript presented in an intelligible fashion and written in standard English?

Reviewer #4: Yes

Reviewer #6: Yes

6. Review Comments to the Author

Reviewer #4: Thanks to the authors for addressing my comments. I believe the manuscript is now in better shape than the previous version. However, the authors can double check the manuscript once more to ensure consistency in terminologies, formatting, and style throughout the text.

Reviewer #6: Dear authors,

Thank you for conducting all the required changes. Now, I consider the paper is ready to be accepted.

Kind regards.

7. PLOS authors have the option to publish the peer review history of their article (what does this mean?). If published, this will include your full peer review and any attached files.

Reviewer #4: No

Reviewer #6: No

---

## [Editor Report · Acceptance letter]

PONE-D-25-15113R1

PLOS ONE

Dear Dr. He,

I'm pleased to inform you that your manuscript has been deemed suitable for publication in PLOS ONE. Congratulations! Your manuscript is now being handed over to our production team.

Kind regards,

on behalf of

Dr. Weicong Li

Academic Editor

PLOS ONE